# IN-CONTEXT CLUSTERING WITH LARGE LANGUAGE MODELS

## ABSTRACT

We propose *In-Context Clustering* (ICC), a flexible LLM-based procedure for clustering data from diverse distributions. Unlike traditional clustering algorithms constrained by predefined similarity measures, ICC flexibly captures complex relationships among inputs through an attention mechanism. We show that pretrained LLMs exhibit impressive zero-shot clustering capabilities on text-encoded numeric data, with attention matrices showing salient cluster patterns. Spectral clustering using attention matrices offers surprisingly competitive performance. We further enhance the clustering capabilities of LLMs on numeric and image data through fine-tuning using the Next Token Prediction (NTP) loss. Moreover, the flexibility of LLM prompting enables text-conditioned image clustering, a capability that classical clustering methods lack. Our work extends in-context learning to an unsupervised setting, showcasing the effectiveness and flexibility of LLMs for clustering.

## 1 INTRODUCTION

Central to any clustering procedure is a similarity measure that makes it possible to separate data into meaningful groups. Classical methods often rely on predefined measures, such as k-means with Euclidean distance, and therefore impose strong assumptions on the underlying data distributions. As a result, these approaches often struggle with high-dimensional and semantically complex data such as text (Liu et al., 2003; Shah & Mahajan, 2012), images (Wazarkar & Keshavamurthy, 2018; Chang et al., 2017; Guérin & Boots, 2018), and audio (Meinedo & Neto, 2003; Alwassel et al., 2020), where similarity is context-dependent and cannot be easily captured by a rigid predefined function.

Recent advances in Large Language Models (LLMs) offer a promising alternative through in-context learning (ICL) (Vaswani et al., 2017; Brown et al., 2020), which has been proven effective across a variety of data distributions (Tsimpoukelli et al., 2021; Garg et al., 2022; Gruver et al., 2023; Vacareanu et al., 2024). Instead of using a predefined similarity function, LLMs capture context-dependent relations through an attention mechanism with query and key projections learned from large-scale pretraining. The ability to recognize contextual relationships among in-context examples provides a foundation for flexible clustering that can adapt to diverse data and different criteria. This LLM-based approach particularly excels in *few-shot scenarios involving semantically rich, naturalistic data*, complementing classical methods optimized for structured large-scale datasets.

In this work, we propose *In-Context Clustering* (ICC), extending in-context learning to an unsupervised setting (Figure 1). Different from previous in-context supervised learning that requires multiple input-output pairs in the prompt (Brown et al., 2020), ICC utilizes only unlabeled input data in the context. Given a natural language instruction specifying the clustering objective and a sequence of inputs, the LLM generates cluster labels autoregressively. When the clustering condition changes (e.g., grouping by color instead of class as shown in Figure 5), one can simply modify the prompt without updating model weights or features. We evaluate ICC on numerical data and image data using a variety of synthetic and real-world datasets to demonstrate the effectiveness and flexibility of ICC.

Our paper is structured as follows:

- We demonstrate that LLMs can provide surprisingly strong zero-shot in-context clustering capabilities (Section 3.1).
- We find attention matrices in intermediate layers show salient cluster structures. Moreover, spectral clustering using these attention matrices yields impressive performance (Section 3.2).

Figure 1: *In-Context Clustering* (ICC). LLMs can flexibly handle diverse modalities and perform text-conditioned clustering. We show the zero-shot clustering capability in pretrained LLMs and further strengthen it through finetuning.

- With lightweight LoRA fine-tuning (Hu et al., 2021) using NTP loss on generated clustering data, we find ICC significantly improves on numeric (Section 4.1) and image data (Section 4.2), especially under heavy-tailed distributions and for images with rich semantics.

- We show that ICC has the relatively distinct ability to do text-conditional image clustering, demonstrating flexibility beyond classical methods. For example, "cluster based on color", or "cluster based on foreground". We believe that this ability to change the way clustering is done based on different prompts makes ICC, and this research direction, particularly compelling. Finally, we show ICC outperforms recent caption-based LLM clustering (Kwon et al., 2024) (Section 5).

## 2 RELATED WORK

**Classical Clustering Algorithms.** Classical clustering methods can be classified into hierarchical, partitional, and density-based methods (Jain et al., 1999; Wazarkar & Keshavamurthy, 2018). Hierarchical methods continuously merge data points into clusters based on their similarity with others, resulting in a dendrogram of the data (Ward Jr, 1963; Murtagh & Contreras, 2012). By contrast, partitional clustering algorithms output a single partition of the data instead of a clustering hierarchy (Ikotun et al., 2023). K-means is one of the most widely used partitional clustering methods based on Euclidean distance and works well for spherical Gaussian clusters. Density-based methods can find arbitrarily shaped clusters by detecting the dense regions in the given dataset (Ester et al., 1996). Although widely used, classical methods lack the ability to do representation learning, instead relying on predefined similarity measures that make strong or often unrealistic assumptions about the data. These drawbacks motivate a more flexible clustering algorithm effective for diverse distributions.

**LLMs for Text Clustering.** LLMs have demonstrated their excellent ability to understand and reason with natural language (Bubeck et al., 2023; Huang & Chang, 2023; Zhang et al., 2024). Recent studies have demonstrated the effectiveness of LLMs in text clustering (Zhang et al., 2023; Viswanathan et al., 2024; Nakshatri et al., 2023; Tipirneni et al., 2024). Various strategies have been explored to enhance clustering performance, including LLM-generated embeddings (Zhang et al., 2023) and few-shot prompting (Viswanathan et al., 2024). However, these practices are limited to text, where the success is somewhat expected, given that the input aligns closely with the pre-training data of the LLMs. In this paper, we extend LLM clustering to non-textual modalities. We find that language pretaining provides a strong foundation for clustering numeric and imagery data.

**Multimodal Clustering.** Multimodal data introduces challenges in aligning heterogeneous information across modalities. Clustering can be performed jointly across modalities using a shared embedding space, or conditionally where one modality guides the clustering of another. As an example for joint multimodal clustering, Su et al. (2024) propose Multimodal Generalized Category Discovery (Multimodal GCD) that focuses on partitioning a shared multimodal embedding space into known and novel categories. As for conditional multimodal clustering, IC|TC (Kwon et al., 2024) and SSD-LLM (Luo et al., 2025) both leverage LLMs for text-conditioned image clustering by converting images to captions. IC|TC distills image captions into one-word labels using an LLM, which are clustered according to the given textual criteria, and the final assignment is made by prompting the

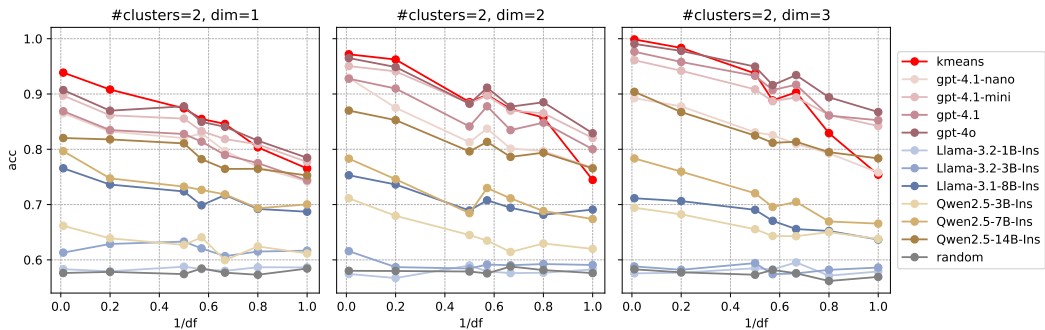

Figure 2: Zero-shot Clustering Accuracy on $t$-Distribution with Different Degrees of Freedom. When $df$ is small, the data distribution has a heavy tail, which violates the Gaussian assumption of k-means. LLMs show impressive zero-shot clustering capabilities on heavy-tailed data.

LLM to match image captions to the cluster labels. SSD-LLM uses LLMs iteratively to refine and produce subpopulation structures based on image captions, and then utilizes the subpopulation structures for clustering. While the task of text-conditioned image clustering is similar to ours in Section 5, these caption-based approaches are highly constrained by the caption quality, failing to generalize when the data has complicated or nuanced relationships that the captioner is unable to capture.

## 3 ZERO-SHOT CLUSTERING

In this section, we show that LLMs pre-trained on large text corpus are capable of zero-shot clustering. LLMs outperform k-means on non-Gaussian data, demonstrating their potential to perform in-context clustering. We also observe that a cluster-like pattern emerges in the self-attention of pretrained LLMs and using the attention matrices for spectral clustering results in competitive performance.

### 3.1 ZERO-SHOT IN-CONTEXT CLUSTERING

**Experimental Setup.** To understand the zero-shot clustering capabilities of different model families and model sizes, we test pre-trained Llama 3.1&3.2 (AI@Meta, 2024), Qwen 2.5 (Bai et al., 2023) with different sizes, and various closed-source GPT models (Achiam et al., 2023) including GPT-4O and GPT-4.1 series. We round all numbers to two decimal places and use text to represent the input numeric data as a double list where the inner list represents one data point. Our prompt is as follows:

> *Cluster the following data into {#clusters} clusters. Only output the cluster labels*
> *for each point as a list of integers. Data: {input data} Labels:*

**Data.** We sample data from a $t$-distribution to evaluate ICC under diverse conditions: When $df$ are large, it approximates the Gaussian distribution; when $df$ are small, it exhibits a heavy tail. We first sample the cluster centroids by drawing each dimension uniformly from $[-10, 10]$, and then generate data points within each cluster by sampling from a $t$-distribution with the specified $df$. For each combination of the number of clusters $c \in \{2, 3, 4, 5\}$, dimensions $d \in \{1, 2, 3, 4\}$, and different degrees of freedom $df \in \{1, 1.25, 1.5, 1.75, 2, 5, 100\}$, we generate 100 samples with length randomly drawn from $[10, 50]$. The size of each cluster is also random but forced to be nonempty.

**Results.** We report zero-shot accuracy[1] in Figure 2 and include more results of different numbers of clusters and dimensions in Figure 6 of Appendix A. LLMs show impressive zero-shot clustering capabilities, outperforming k-means when the data has heavy tails. When $df$ is small, the Gaussian assumption of k-means is violated, leading to a significant drop in performance. GPT-4 and GPT-4.1 outperform k-means when data is heavy-tailed and high-dimensional, demonstrating the potential of applying LLMs for clustering high-dimensional non-Gaussian data.

---

[1]Since clustering is invariant to label permutation, we adopt the Hungarian Algorithm to find the optimal assignment before computing the accuracy.

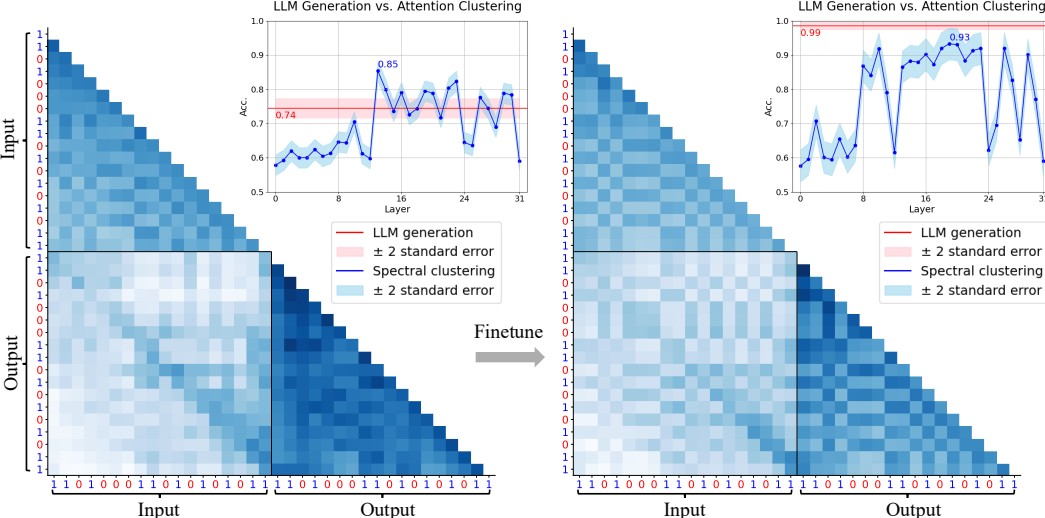

Figure 3: Visualization of Attention Allocation of Input Data and Generated Cluster Labels at an Intermediate Layer. The x-axis and y-axis are the ground-truth cluster labels. The left figure is for the pretrained LLAMA-3.1-8B-INSTRUCT, and the right is after fine-tuning(details in Section 4.1). The top right curves are the average accuracy of spectral clustering using the input-input attention score matrices (top-left) across different layers, compared with the average accuracy of LLM generation.

The performance of LLMs is correlated with the model size and training choices. Small LLMs with 3B or 8B parameters can produce non-trivial answers when the clustering data is simple (with lower dimensions and fewer clusters, shown in Figure 6). When the data becomes more complicated, these small LLMs are either unable to follow the instruction of generating the correct number of clusters or produce answers that are close to random guesses. We also observe that instruction tuning improves the overall accuracy, without which the model is unable to follow the instructions of the clustering task (Figure 7). There is still a gap between the performance of small open-source models and GPT models, probably due to the difference in the model size and pretraining. In Section 4, we show that finetuning Llama models on synthetic clustering data helps close the gap.

## 3.2 EMERGENCE OF CLUSTERS IN ATTENTION

To better understand the inner mechanism of ICC, we visualize the attention scores across different transformer layers. All LLMs considered here are causal transformers with multi-head self-attention. Given a textual prompt as described in Section 3, the model autoregressively generates cluster labels conditioned on the input data and previous generation. At each layer, we extract the self-attention matrix $A \in \mathbb{R}^{n \times n}$, a lower-triangular matrix due to causality, where $n$ is the total number of tokens. For multi-head attention, we use average attention scores across heads in this section.

To focus on input data and output cluster label tokens, we discard instruction and system prompt tokens. Since each input data point may span multiple tokens, we aggregate token-level attention scores to obtain data-level attention scores. Let $m$ denote the number of input data points. From the full matrix $A$, we construct an aggregated attention matrix with the following block structure:

$$A = \begin{bmatrix} A^{II} & 0 \\ A^{OI} & A^{OO} \end{bmatrix}. \tag{1}$$

Here, $A^{II} \in \mathbb{R}^{m \times m}$ represents the input-input matrix capturing attention scores among input data points, $A^{OI} \in \mathbb{R}^{m \times m}$ represents the output-input matrix that reflects how generated cluster labels attend to input data, and $A^{OO} \in \mathbb{R}^{m \times m}$ represents the output-output matrix containing attention scores among output tokens. Each input data point $d_i$ may span multiple tokens, indexed from $s_i$ to $e_i$. We compute $A^{II}$ by averaging attention scores across all token pairs between $d_i$ and $d_j$:

$$A_{ij}^{II} := \frac{1}{(e_i - s_i + 1)(e_j - s_j + 1)} \sum_{p=s_i}^{e_i} \sum_{q=s_j}^{e_j} A_{pq}. \tag{2}$$

Each output cluster label is represented by a single token, indexed as $t_i$ for the label of $d_i$. The remaining attention blocks are defined as:

$$A_{ij}^{OI} := \frac{1}{e_j - s_j + 1} \sum_{p=s_j}^{e_j} A_{t_i p}, \quad A_{ij}^{OO} := A_{t_i t_j}. \tag{3}$$

Figure 3 visualizes this block matrix , with $A^{II}$ in the top-left, $A^{OI}$ in the bottom-left, and $A^{OO}$ in the bottom-right. Here, we take one clustering example generated from Gaussian distribution with two clusters. We observe that *attention matrices in intermediate layers show block structures that align with cluster identities*. The transformer assigns higher attention scores to similar data within the same cluster that has been seen in the past. We provide more examples across different layers in Appendix B.1. This cluster pattern is consistent and salient in most middle layers. In contrast, the final layer typically shows a vertical-slash pattern, as also observed by Jiang et al. (2024). We also observe that most attention heads show similar cluster patterns in Figure 10.

Although the pretrained model (left in Figure 3) has a clear cluster pattern in the input-input matrix, clusters are not observed in attention related to outputs. This suggests that the model learns similarity among input data during pretraining, but is not optimized for generating cluster labels as explicit clustering tasks are very likely rare in pretraining.[2] After fine-tuning on ICC data, the cluster structure in the input-input matrix becomes stronger, and similar clusters also emerge in output-input and output-output matrices.

To quantify how well the attention captures the similarity among the input data, we use these input-input attention score matrices for spectral clustering (Ng et al., 2001; von Luxburg, 2007) (more details and results are in Appendix B.2). Although the zero-shot accuracy of prompting pretrained LLAMA-3.1-8B-INSTRUCT to cluster is 74%, the spectral clustering using attention with the optimal choice of layers achieves 85% before fine-tuning. This surprising result suggests that attention of LLMs already encodes rich structural information beyond what is directly generated. In addition to prompting the LLM for generation, directly using attention can be an alternative to leverage pretrained LLM for in-context clustering in zero shot.

## 4 LEARNING CLUSTERING WITH NEXT TOKEN PREDICTION

While pretrained LLMs show promising zero-shot clustering capabilities, small open-source models lag behind classical methods and proprietary LLMs. In this section, we show that the clusterng capabilities of pretrained LLMs can be further enhanced through LoRA fine-tuning using NTP loss. Inspired by the meta learning literature (Ravi & Larochelle, 2017; Min et al., 2022; Najdenkoska et al., 2023), we construct various clustering episodes to make pretrained (multimodal) LLM learn to cluster in context and then test it on unseen classes. We experiment on both numeric and image data.

### 4.1 NUMERIC DATA CLUSTERING

**Experiment Setup.** We follow the standard Supervised Fine-Tuning (SFT) procedure to fine-tune pre-trained Llama models with different sizes (LLAMA-3.2-1B-INSTRUCT, LLAMA-3.2-3B-INSTRUCT, LLAMA-3.1-8B-INSTRUCT) using NTP loss. Similarly to how we construct the clustering data in Section 3, we construct the data by randomly sampling data from a $t$-distribution with different degrees of freedom $df \in \{1, 2, 5, 100\}$, the number of clusters $c \in \{2, 3, 4, 5\}$, and dimensions of each point $d \in \{1, 2, 3, 4\}$. We generate around 100k input-label pairs, where each sample has a length randomly drawn from $[10, 50]$. We use LoRA (Hu et al., 2021) to fine-tune the pre-trained Llama model for one epoch with an effective batch size of 32 and a learning rate of 5e-4.

**Results.** We use the test data in Section 3 ($df \in \{1, 1.25, 1.5, 1.75, 2, 5, 100\}$) with $df \in \{1.25, 1.5, 1.75\}$ to test the robustness of the fine-tuned model. During fine-tuning, the LLM exhibits a two-phase learning pattern where it first learns the correct format and then gradually develops a clustering mechanism. Initially, the LLM (especially smaller models with 1B or 3B parameters) struggles with instruction following and produces repetitive outputs. These poorly formatted predictions are heavily penalized by the NTP loss. As training progresses, the model learns to effectively differentiate among cluster labels based on the input data and achieves a high accuracy.

---

[2]Llama 3 models are claimed to be trained on "15T tokens that were all collected from publicly available sources"(AI@Meta, 2024), but details are not disclosed.

Table 1: Effect of Finetuning on $t$-Distributed Data with Different Degrees of Freedom. Input $dim = 3$ and number of clusters $c = 3$. We report average accuracy (%) and one standard error.

| | DF=1 | DF=1.25 | DF=1.5 | DF=1.75 | DF=2 | DF=5 | DF=100 |
|---|---|---|---|---|---|---|---|
| KMEANS | $67.95_{\pm1.46}$ | $75.43_{\pm1.52}$ | $85.57_{\pm1.20}$ | $87.55_{\pm1.32}$ | $89.05_{\pm1.27}$ | $95.29_{\pm1.00}$ | $97.08_{\pm0.82}$ |
| GPT-4O | $77.75_{\pm1.31}$ | $80.60_{\pm1.20}$ | $86.99_{\pm1.15}$ | $87.08_{\pm1.26}$ | $89.56_{\pm1.10}$ | $93.84_{\pm1.03}$ | $96.25_{\pm0.86}$ |
| (A) LLAMA-3.2-1B-INSTRUCT | $45.40_{\pm0.64}$ | $47.09_{\pm0.71}$ | $46.77_{\pm0.66}$ | $46.63_{\pm0.67}$ | $46.54_{\pm0.69}$ | $45.73_{\pm0.64}$ | $47.36_{\pm0.77}$ |
| (A) + FINETUNE | $82.66_{\pm1.30}$ | $86.45_{\pm1.23}$ | $91.10_{\pm0.90}$ | $89.46_{\pm1.18}$ | $88.76_{\pm1.20}$ | $95.09_{\pm0.93}$ | $96.28_{\pm0.88}$ |
| (B) LLAMA-3.2-3B-INSTRUCT | $46.71_{\pm0.67}$ | $46.09_{\pm0.72}$ | $46.35_{\pm0.62}$ | $46.85_{\pm0.76}$ | $46.05_{\pm0.82}$ | $46.84_{\pm0.72}$ | $46.35_{\pm0.86}$ |
| (B) + FINETUNE | $88.54_{\pm1.03}$ | $91.05_{\pm1.00}$ | $94.31_{\pm0.77}$ | $93.33_{\pm0.90}$ | $94.51_{\pm0.90}$ | $98.08_{\pm0.49}$ | $97.64_{\pm0.78}$ |
| (C) LLAMA-3.1-8B-INSTRUCT | $55.29_{\pm1.34}$ | $55.38_{\pm1.44}$ | $59.80_{\pm1.57}$ | $61.09_{\pm1.55}$ | $61.21_{\pm1.47}$ | $64.73_{\pm1.66}$ | $64.42_{\pm1.73}$ |
| (C) + FINETUNE | $\mathbf{90.66}_{\pm0.95}$ | $\mathbf{92.20}_{\pm0.93}$ | $\mathbf{95.25}_{\pm0.54}$ | $\mathbf{94.57}_{\pm0.86}$ | $\mathbf{95.44}_{\pm0.71}$ | $\mathbf{98.90}_{\pm0.31}$ | $\mathbf{97.85}_{\pm0.76}$ |

As shown in Table 1, all fine-tuned models show superior performance compared to k-means and GPT-4O (the complete results are in Figure 7 of Appendix A). Although these LLMs are fine-tuned on $t$-distributed data with $df \in \{1, 2, 5, 100\}$, they show generalization capability to more $df$ and different distributions. All fine-tuned models perform consistently well on $t$-distributed data with new $df \in \{1.25, 1.5, 1.75\}$. While these models are fine-tuned on a symmetric distribution, they also significantly outperform k-means and GPT-4O on a skewed distribution (lognormal) as shown in Table 4 in Appendix A. We also observe that models with higher accuracy tend to be more invariant to permutation in input data, and data augmentation is effective in improving consistency, as shown in Table 5.

We study the effect of fine-tuning by analyzing the attention pattern as visualized in Figure 3. The cluster pattern in the attention score matrix of the input data is significantly more salient after fine-tuning, indicating that the model learns a better similarity function among the data through its attention mechanism during fine-tuning. The accuracy of spectral clustering using attention scores increases as well. More visualization and results are in Appendix B.

## 4.2 IMAGE CLUSTERING

Here, we extend ICC to multimodal LLMs and present results of image clustering. Given a set of images, the goal is to cluster based on their semantic meanings. By projecting image embeddings obtained from a pretrained visual encoder, LLMs can learn to produce meaningful groupings that outperform an LLM-based method that relies on image captions.

**Model.** We use `llava-interleave-qwen-7b-hf` (Li et al., 2024a), a multimodal LLM pretrained with multi-image inputs, as our base model. In the LLaVA framework, each image is segmented into 729 patches encoded by a pre-trained ViT, namely the SigLIP's visual encoder (Zhai et al., 2023), then projected through an MLP layer into the embedding space of the base LLM (Bai et al., 2023). While such a high-granularity representation may benefit downstream tasks like object detection, we argue that it is not optimal for clustering tasks. Clustering typically involves a large number of images; thus, using hundreds of tokens per image can quickly exceed context length limitations and significantly increase computational costs during fine-tuning. Additionally, high granularity might be unnecessary for some clustering tasks that only rely on global features.

To address these efficiency concerns, we implement average pooling after the projection layer to reduce per-image token lengths, as illustrated in Figure 4 (left). Each input image is divided into patches, which are preprocessed and flattened (omitted from the figure for clarity), and then encoded by a vision transformer. We reshape the flattened image features back to 2D and then apply average pooling to reduce dimensionality. The pooled features are then flattened, projected into the LLM's embedding space, and concatenated with text token embeddings. We experiment with various pooling kernel sizes in Appendix C.1. No padding is applied and the stride is the same as the kernel width.

**Data.** We collect images from ImageNet21k (Ridnik et al., 2021) where images sharing the same label are considered part of the same cluster. We reserve the 384 image classes covered in ImageNet-with-Attributes (Russakovsky & Fei-Fei, 2010) for testing and the remaining 18K classes for training. For training, we construct 192K image clustering episodes of various numbers of clusters $c \in \{2, 3, 4\}$, with random length $l \in [10, 30]$ and random cluster proportion. For testing, we use the reserved test classes to construct 100 clustering episodes for each number of clusters. To test generalization on out-of-domain data, we include Plant Disease and EuroSAT datasets from the Cross-Domain Few-Shot Learning (CD-FSL) Benchmark (Guo et al., 2020) with details in Appendix C.2.

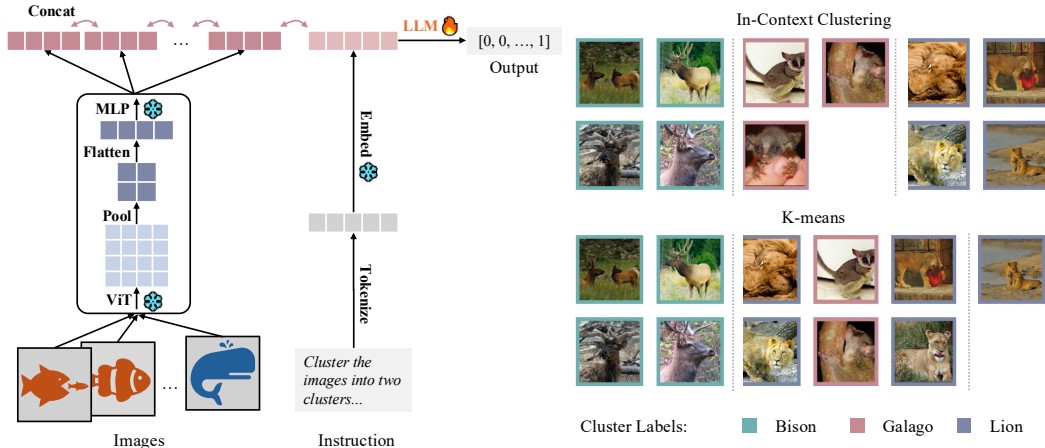

Figure 4: Left: Multimodal LLM Architecture with Average Pooling for Image Features. Right: Qualitative Comparison of Models on Image Clustering — ICC outperforms k-means when the data has rich semantic information.

Table 2: Image Clustering Accuracy (%) with Standard Error. ICC(GPT-4O) is zero-shot ICC using gpt-4o and the shaded rows represent models finetuned on ImageNet data with numbers of clusters $c \in \{2, 3, 4\}$, where SMALL, MEDIUM, LARGE refer to the per-image token length in Appendix C.1. Our finetuned models can generalize to unseen $c = 5$ and other datasets that deviate from ImageNet.

| NUMBER OF CLUSTERS | IMAGENET | | | | PLANT | EUROSAT |
| | c=2 | c=3 | c=4 | c=5 | c=2 | c=2 |
| --- | --- | --- | --- | --- | --- | --- |
| K-MEANS | $89.43_{\pm1.57}$ | $82.09_{\pm1.44}$ | $79.07_{\pm1.31}$ | $77.96_{\pm1.08}$ | $\mathbf{93.70}_{\pm1.40}$ | $\mathbf{85.52}_{\pm1.43}$ |
| IC\|TC(KWON ET AL., 2024) | $90.20_{\pm1.54}$ | $78.86_{\pm1.41}$ | $76.49_{\pm1.50}$ | $73.99_{\pm1.58}$ | $67.40_{\pm1.23}$ | $72.97_{\pm1.42}$ |
| ICC (GPT-4O) | $82.46_{\pm1.40}$ | $80.25_{\pm1.73}$ | $75.91_{\pm1.73}$ | $78.08_{\pm1.50}$ | $84.74_{\pm1.25}$ | $\underline{79.08}_{\pm1.41}$ |
| ICC (SMALL) | $96.81_{\pm0.83}$ | $91.94_{\pm1.03}$ | $89.83_{\pm1.19}$ | $82.08_{\pm1.01}$ | $73.03_{\pm1.58}$ | $78.17_{\pm1.53}$ |
| ICC (MEDIUM) | $\underline{98.26}_{\pm0.71}$ | $\mathbf{95.92}_{\pm0.90}$ | $\underline{91.62}_{\pm1.16}$ | $\underline{84.92}_{\pm0.95}$ | $82.28_{\pm1.85}$ | $78.64_{\pm1.61}$ |
| ICC (LARGE) | $\mathbf{99.12}_{\pm0.41}$ | $\underline{91.95}_{\pm0.96}$ | $\mathbf{92.92}_{\pm1.06}$ | $\mathbf{84.96}_{\pm0.89}$ | $\underline{85.09}_{\pm1.80}$ | $77.35_{\pm1.70}$ |

**Experiment Setup.** Similarly to previous numerical experiments, we use LoRA to fine-tune the LLM with NTP loss. The visual encoder and projection layer are frozen during training. We fine-tune for one epoch with an effective batch size of 32 and a learning rate of 5e-4.

**Baselines.** To ensure a fair comparison, we use average-pooled image features from the vision encoder of the base model (Li et al., 2024a) as the inputs to k-means. We also compare ICC against IC\|TC (Kwon et al., 2024), a recent LLM-based image clustering method. We use the same model (Li et al., 2024a) to generate image captions for IC\|TC then use GPT-3.5-TURBO to distill and cluster the captions according to the given number of clusters and the clustering condition. Although converting images to short captions facilitates clustering via LLMs, IC\|TC experiences information loss during the captioning and summarization stage, limiting its performance on challenging data.

**Results.** The performance of different models is summarized in Table 2. While zero-shot ICC using GPT-4O achieves competitive performance, it is less effective than on text-encoded data. This is likely due to the current limitations of multimodal LLMs on long sequences of complex images. Our proposed finetuning method significantly closes this gap, achieving strong performance across all datasets. Despite being only fine-tuned on ImageNet data with the number of clusters less than five, our model can generalize to within-domain data of five clusters and out-of-domain data including plant leaves and satellite images.

With good image features, k-means is effective on datasets with limited semantic complexity, such as Plant Disease and EuroSAT. However, it loses its competence on ImageNet, where images often depict complex scenes involving multiple objects. The caption-based method, IC\|TC, performs poorly on Plant Disease or EuroSAT, as its captioning model lacks domain-specific knowledge. This observation highlights a key weakness of caption-based clustering: its dependence on accurate and relevant captions limits its applicability to novel domains. Our model avoids these pitfalls, demonstrating superior flexibility and performance across both general and specialized domains.

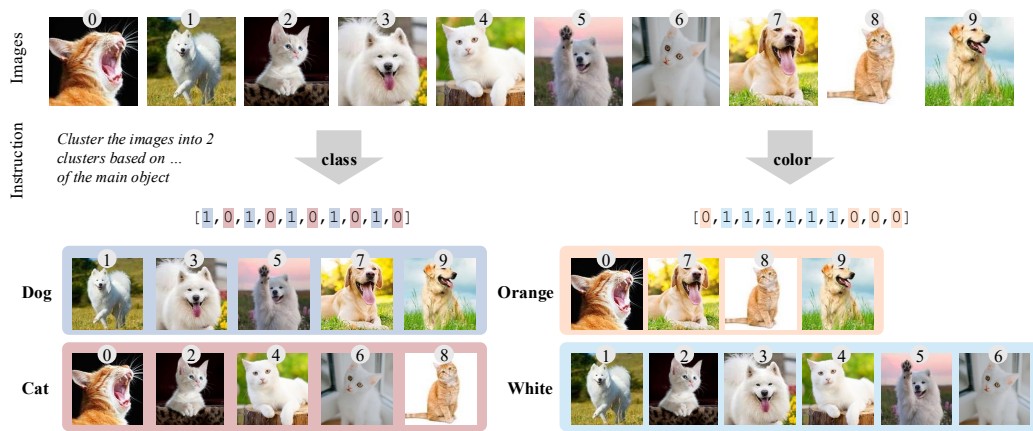

Figure 5: LLMs are able to produce different clusterings according to the condition in the prompt.

## 5  TEXT-CONDITIONED CLUSTERING

While the experiments in the previous section assume a single, fixed clustering objective, real-world data admits multiple plausible clusterings depending on the objective. For example, the same set of animal images can be clustered by visual properties like colors (orange vs. white) or semantic categories like species (dog vs. cat), as shown in Figure 5. When the clustering condition changes, classical methods typically require retraining or re-engineering features. In contrast, LLMs can easily adapt to new conditions through prompting thanks to their powerful contextual understanding capability. In this section, we perform text-conditioned image clustering by fine-tuning multimodal LLMs with the NTP loss.

**Data.** We construct conditional clustering using ImageNet-with-Attributes (Russakovsky & Fei-Fei, 2010), which includes 384 classes with 4 categories of attributes (COLOR, SHAPE, PATTERN, TEXTURE). We split the data into 80% training classes and 20% testing classes. We treat the category name as the clustering condition that will be specified in the prompt and use the attribute value as cluster labels. In addition, we include an OBJECT category that is similar to Section 4.2, where we use the class name of the images as cluster labels. Images with ambiguous annotations are filtered out. For training, we construct around 280K image conditional clustering episodes of various numbers of clusters $c \in \{2, 3, 4\}$,[3] with random length $l \in [10, 30]$ and random cluster proportion.

To test the performance of the model on different conditions, we use the reserved test classes of ImageNet-with-Attributes and also include the Stanford 40 Action dataset (Yao et al., 2011) with annotations on the LOCATION of the scene, the ACTION and MOOD of the people in the image provided by (Kwon et al., 2024). For each dataset and clustering condition, we sample 100 clustering data from two random classes of each attribute category, with random size $l \in [10, 30]$ and random cluster proportion.

**Experiment Setup.** Following the SFT procedure in Section 4.2, we use LoRA to fine-tune `llava-interleave-qwen-7b-hf` with different pooling ratios. We keep the visual encoder and projection layer frozen during training. We use NTP loss to fine-tune for one epoch with an effective batch size of 32 and a learning rate of 5e-4.

**Baselines.** We test both unconditional and conditional clustering methods. K-means is a unconditional baseline as it does not allow injecting clustering criteria. For conditional clustering methods, we test IC|TC explicitly specifying conditions in the prompts for all the summarization and clustering stages, with GPT-3.5-TURBO as the LLM to save costs.

---

[3]The pattern category only has two available values, so we don't have $c \in \{2, 3\}$ for this category.

Table 3: Conditional Image Clustering Accuracy (%) with Standard Error. Here, ICC (MEDIUM:4.2) represents the model finetuned on unconditional image clustering data in Section 4.2, while others use conditional image clustering data in Section 5. Our method outperforms all baselines on ImageNet and Stanford 40 Action. SMALL, MEDIAN, LARGE refer to the per-image token length in Appendix C.1.

| | | | IMAGENET | | | STANFORD 40 ACTION | | |
|---|---|---|---|---|---|---|---|---|
| | OBJECT | COLOR | PATTERN | SHAPE | TEXTURE | ACTION | MOOD | LOCATION |
| *Unconditional Methods* | | | | | | | | |
| K-MEANS | $89.96_{\pm1.44}$ | $66.40_{\pm1.16}$ | $62.36_{\pm0.98}$ | $75.76_{\pm1.78}$ | $78.53_{\pm1.65}$ | $79.90_{\pm1.76}$ | $70.93_{\pm1.43}$ | $78.11_{\pm1.50}$ |
| *Conditional Methods* | | | | | | | | |
| IC\|TC(KWON ET AL., 2024) | $91.93_{\pm1.38}$ | $69.70_{\pm1.35}$ | $76.12_{\pm1.53}$ | $70.15_{\pm1.34}$ | $68.74_{\pm1.34}$ | $93.74_{\pm1.25}$ | $75.65_{\pm1.35}$ | $75.49_{\pm1.64}$ |
| ICC(GPT-4O) | $67.58_{\pm1.30}$ | $66.36_{\pm1.22}$ | $65.61_{\pm1.12}$ | $70.15_{\pm1.72}$ | $73.54_{\pm1.54}$ | $80.59_{\pm1.28}$ | $68.61_{\pm1.61}$ | $67.75_{\pm1.33}$ |
| ICC (SMALL) | $98.25_{\pm0.71}$ | $76.31_{\pm1.38}$ | $85.50_{\pm0.78}$ | $81.75_{\pm1.69}$ | $82.82_{\pm1.62}$ | $89.60_{\pm1.52}$ | $67.89_{\pm1.27}$ | $\underline{83.84}_{\pm1.53}$ |
| ICC (MEDIUM) | $98.64_{\pm0.58}$ | $\underline{81.02}_{\pm1.31}$ | $\underline{93.28}_{\pm0.56}$ | $\underline{83.02}_{\pm1.69}$ | $\underline{86.04}_{\pm1.52}$ | $\mathbf{95.98}_{\pm1.04}$ | $\underline{76.77}_{\pm1.39}$ | $77.18_{\pm1.67}$ |
| ICC (MEDIUM:4.2) | $\underline{98.88}_{\pm0.55}$ | $71.39_{\pm1.31}$ | $65.04_{\pm1.01}$ | $72.72_{\pm1.37}$ | $83.04_{\pm1.55}$ | $\mathbf{96.47}_{\pm0.95}$ | $\mathbf{78.46}_{\pm1.46}$ | $\mathbf{86.19}_{\pm1.53}$ |
| ICC (LARGE) | $\mathbf{99.52}_{\pm0.22}$ | $\mathbf{84.29}_{\pm1.26}$ | $\mathbf{94.43}_{\pm0.40}$ | $\mathbf{83.72}_{\pm1.71}$ | $\mathbf{87.27}_{\pm1.44}$ | $94.14_{\pm1.26}$ | $73.42_{\pm1.47}$ | $81.72_{\pm1.62}$ |

**Results.**    The quantitative evaluation of different models is summarized in Table 3 and qualitative examples are shown in Appendix D. Similar to results in Section 4.2, zero-shot performance of GPT-4O is promising but ultimately falls short of our finetuned approach. Our finetuned models outperform all baselines on ImageNet and Stanford 40 Action. In general, our method with higher per-image token lengths performs better in this conditional clustering task. Unlike experiments in Section 4.2 where the difference between different granularity is small, this task requires more fine-grained information and thus using more tokens to represent images is preferred. K-means and caption-based IC\|TC often fail to capture such details, particularly for attributes like COLOR, SHAPE, and PATTERN, where our method is more than 10% higher than all baselines.

Our method generalizes to unseen data and conditions from the Stanford 40 Action dataset. Surprisingly, our model trained solely on clustering objects in ImageNet, achieves the highest accuracy. This suggests that the inductive bias from image-based clustering and the visual-language pretraining enables the model to infer clustering objectives implicitly. We notice that the finetuned models are less competitive on MOOD and LOCATION. We attribute this to the training data (ImageNet-with-Attributes), which emphasizes prominent foreground objects (typically non-human), causing the model to overlook cues from human facial expressions or the background. Scaling our approach to more diverse datasets and clustering conditions could mitigate this bias and further strengthen the model's generalization capabilities.

# 6    CONCLUSION

In-Context Clustering (ICC) generalizes in-context learning to the unsupervised setting. ICC does not make restrictive assumptions on the input data and enables flexible, text-conditioned clustering objectives through prompting. We find that large LLMs exhibit strong zero-shot performance on text-encoded numeric data, and further show that this capability can be significantly strengthened for smaller and multimodal models through simple fine-tuning using the NTP loss. Multimodal LLMs enhanced by our proposed finetuning achieve impressive performance on image clustering and text-conditioned image clustering. These findings highlight that LLMs can be effectively used to solve clustering tasks that involve complex semantics and contextual understanding.

While we demonstrate ICC's effectiveness and flexibility, ICC is complementary to classical clustering methods, and has certain limitations. For application to larger datasets, it would be particularly promising to scale ICC to longer contexts, which can be computationally expensive for LLMs (Li et al., 2024b; Liu et al., 2024). Our experiments with average pooling for image features show promise in reducing token usage, and recent advances such as dynamic context selection (Hao et al., 2025) and token pruning (Chen et al., 2024; Jianjian et al., 2024) can further address the long-context challenge in future work. Moreover, while visualizing attention provides some insights into the way ICC performs clustering, a theoretical understanding of ICC would be particularly valuable. Emergence of clusters in self-attention have been theoretically studied by Geshkovski et al. (2023), but under a simplified setting (without multi-head attention, feed-forward layers, and layer normalization). Developing theoretical frameworks to explain and exploit these attention structures remains an important open direction.

## 7 REPRODUCIBILITY STATEMENT

We provide the code for our experiments at `https://anonymous.4open.science/r/ICC-B4CA`. We have included necessary details for reproducing our results in this paper.

- Numeric Data Clustering: The experimental setup for zero-shot experiments, including the data generation process and prompts, is detailed in Section 3 [Experimental Setup & Data]. The fine-tuning procedure is described in Section 4.1 [Experimental Setup]. Additional implementation details are listed in Appendix E.1.

- Image Clustering: The model architecture is described in Section 4.2 [Model]. Data and experiment setup for unconditional and text-conditioned image clustering is in Section 4.2 [Model & Data] and Section 5 [Experimental Setup & Data] respectively. Additional implementation details are provided in Appendix E.2.

- Attention Analysis: Details for processing attention matrices for visualization and spectral clustering is explained in Section 3.2 and Appendix B.

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

# Appendix

## A ADDITIONAL RESULTS OF NUMERIC DATA CLUSTERING

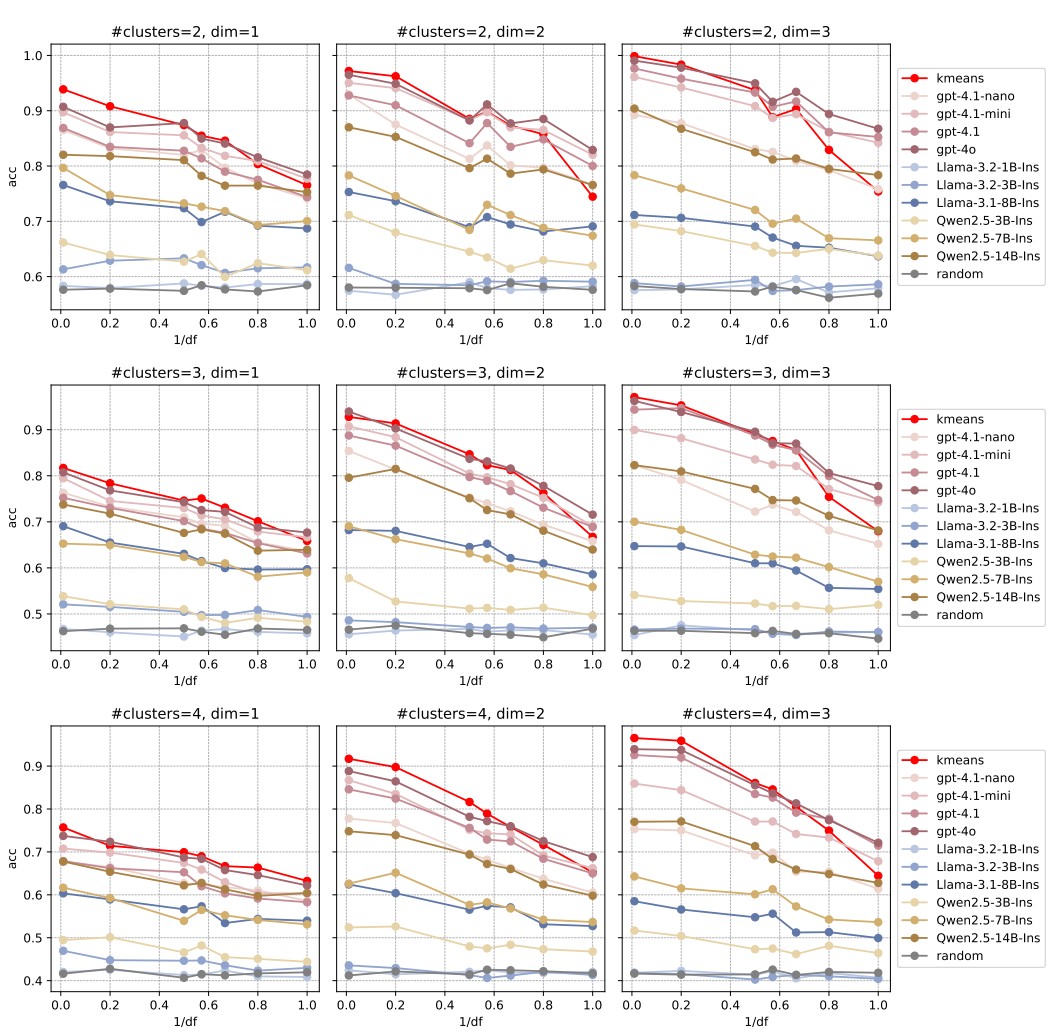

Figure 6: Zero-shot Clustering Accuracy. Test data is t-distributed with different degrees of freedom, number of clusters and dimensions. Note that "Ins" represents "Instruct" in the legend.

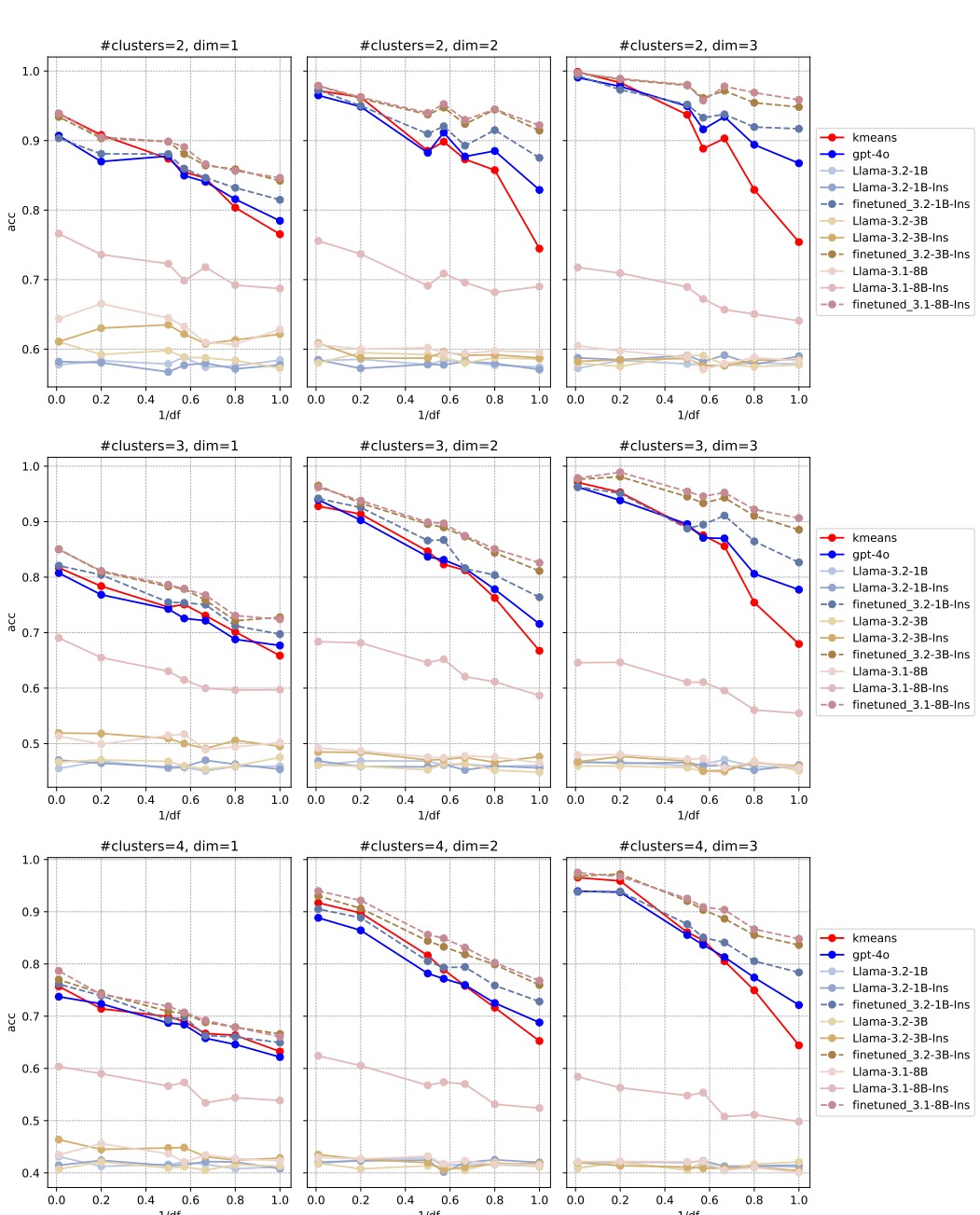

Figure 7: Impact of Instruction Tuning and Clustering-Specific Fine-tuning on Clustering Accuracy. Test data is t-distributed with different degrees of freedom, number of clusters and dimensions. Note that "Ins" represents "Instruct", and "finetune" refers to the fine-tuning on t-distributed clustering data with $df \in \{1, 2, 5, 100\}$ as in Section 4.1.

Table 4: Average Clustering Accuracy with One Standard Error on Lognormal Data. FINETUNED represents the fine-tuned LLAMA-3.1-8B model on t-distributed clustering data with $df \in \{1, 2, 5, 100\}$ as in Section 4.1. Although the model is not fine-tuned on lognormal data, it still outperforms other models in almost all settings.

|  |  | $c = 2$ | $c = 3$ | $c = 4$ |
|---|---|---|---|---|
| | KMEANS | $0.86_{\pm 0.03}$ | $0.77_{\pm 0.02}$ | $0.74_{\pm 0.02}$ |
| $dim = 1$ | GPT-4O | $0.87_{\pm 0.02}$ | $0.75_{\pm 0.02}$ | $0.73_{\pm 0.02}$ |
| | FINETUNED | $\mathbf{0.89}_{\pm 0.02}$ | $\mathbf{0.79}_{\pm 0.02}$ | $\mathbf{0.76}_{\pm 0.02}$ |
| | KMEANS | $0.91_{\pm 0.03}$ | $0.87_{\pm 0.02}$ | $0.82_{\pm 0.02}$ |
| $dim = 2$ | GPT-4O | $0.91_{\pm 0.02}$ | $0.84_{\pm 0.02}$ | $0.80_{\pm 0.02}$ |
| | FINETUNED | $\mathbf{0.94}_{\pm 0.02}$ | $\mathbf{0.91}_{\pm 0.02}$ | $\mathbf{0.86}_{\pm 0.02}$ |
| | KMEANS | $\mathbf{0.98}_{\pm 0.01}$ | $0.92_{\pm 0.02}$ | $0.91_{\pm 0.02}$ |
| $dim = 3$ | GPT-4O | $0.94_{\pm 0.01}$ | $0.86_{\pm 0.02}$ | $0.88_{\pm 0.02}$ |
| | FINETUNED | $0.94_{\pm 0.02}$ | $\mathbf{0.94}_{\pm 0.02}$ | $\mathbf{0.92}_{\pm 0.02}$ |

Table 5: Sensitivity to Input Order. The reported values are average accuracy on t-distributed (c=2, dim=3) data, with average standard deviation over five runs of permuted input data in parentheses. We use the standard deviation to reflect the consistency of clustering methods given permutations of input data. FINETUNED denotes the LLAMA-3.1-8B model finetuned on t-distributed clustering data in Section 4.1, and FINETUNED-AUG denotes finetuning on augmented data with 3 times of permutation. We notice that the model with higher clustering accuracy tends to be more invariant to permutation in input data. Data augmentation is also effective in improving the consistency.

|  | DF=1 | DF=2 | DF=5 | DF=100 |
|---|---|---|---|---|
| K-MEANS | 0.75(0.04) | 0.95(0.03) | **0.99(0.00)** | **0.99(0.00)** |
| GPT-4O | 0.83(0.08) | 0.95(0.03) | 0.97(0.02) | 0.98(0.01) |
| FINETUNED | 0.92(0.04) | 0.97(0.02) | 0.98(0.01) | 0.99(0.01) |
| FINETUNED-AUG | **0.93(0.03)** | **0.98(0.01)** | 0.98(0.01) | **0.99(0.00)** |

# B  EMERGENCE OF CLUSTERS IN ATTENTION

## B.1  ATTENTION OF DIFFERENT LAYERS AND ATTENTION HEADS

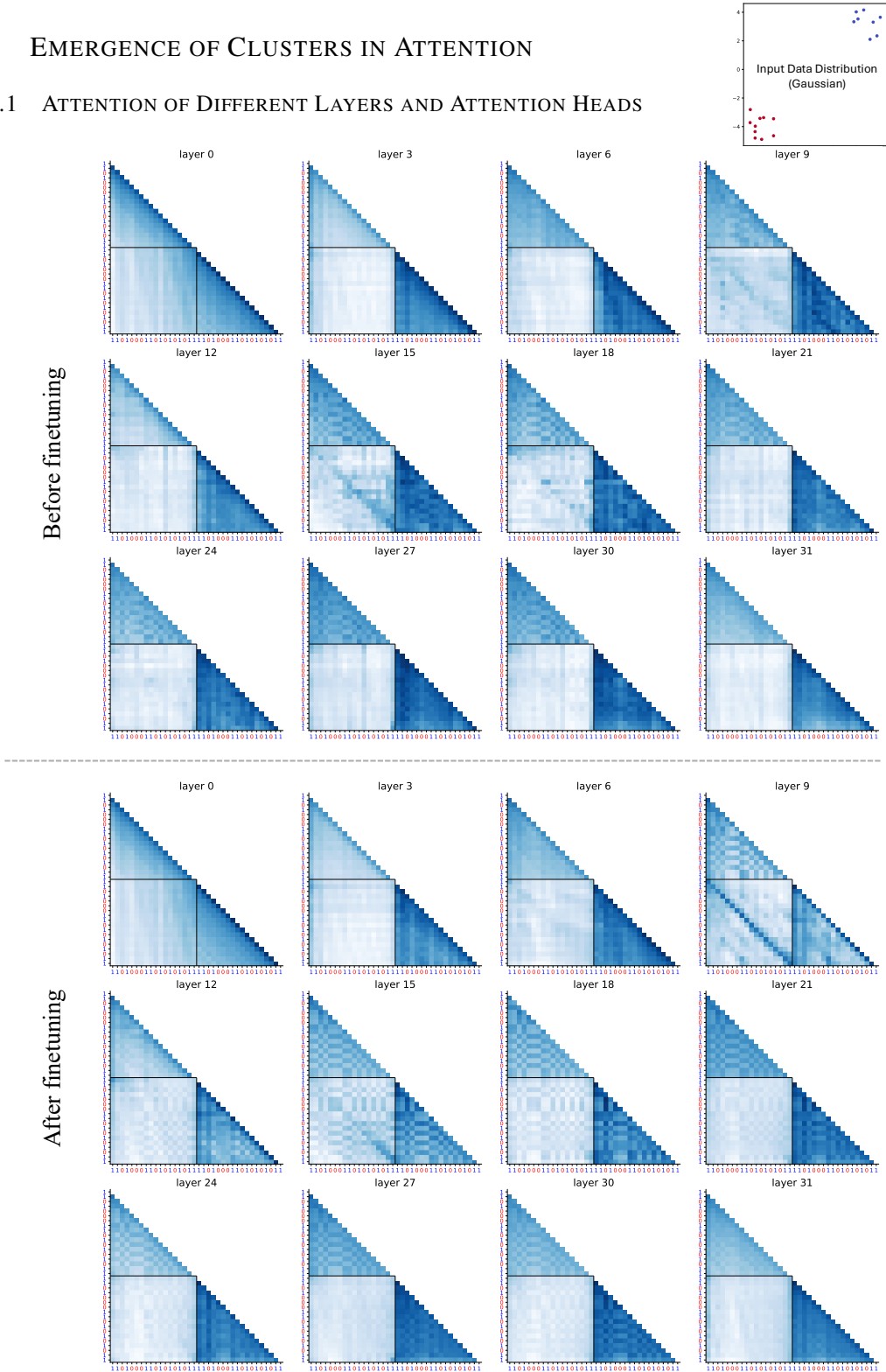

Figure 8: Attention Allocation of LLAMA-3.1-8B-INSTRUCT across Layers. The attention scores are logarithmized for better visualization. Each cluster is generated from a Gaussian distribution, as shown in top right. Figure 3 is a zoom-in view of layer 15 here.

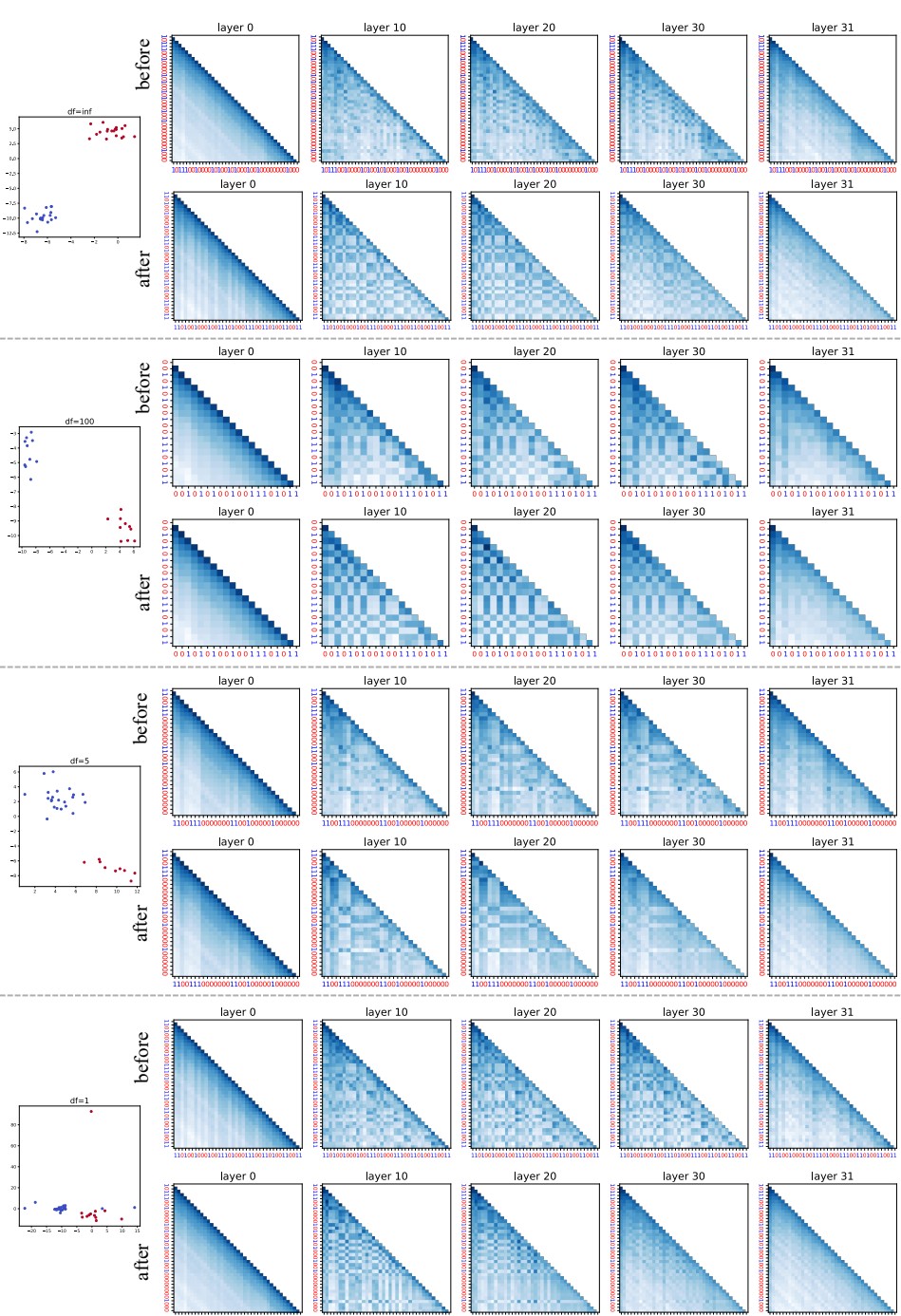

Figure 9: Attention Allocation of LLAMA-3.1-8B-INSTRUCT on $t$-Distributed Data with Different $df$, before and after Finetuning. Note that $t$-distribution with $df = inf$ is Gaussian. The attention scores are logarithmized for better visualization.

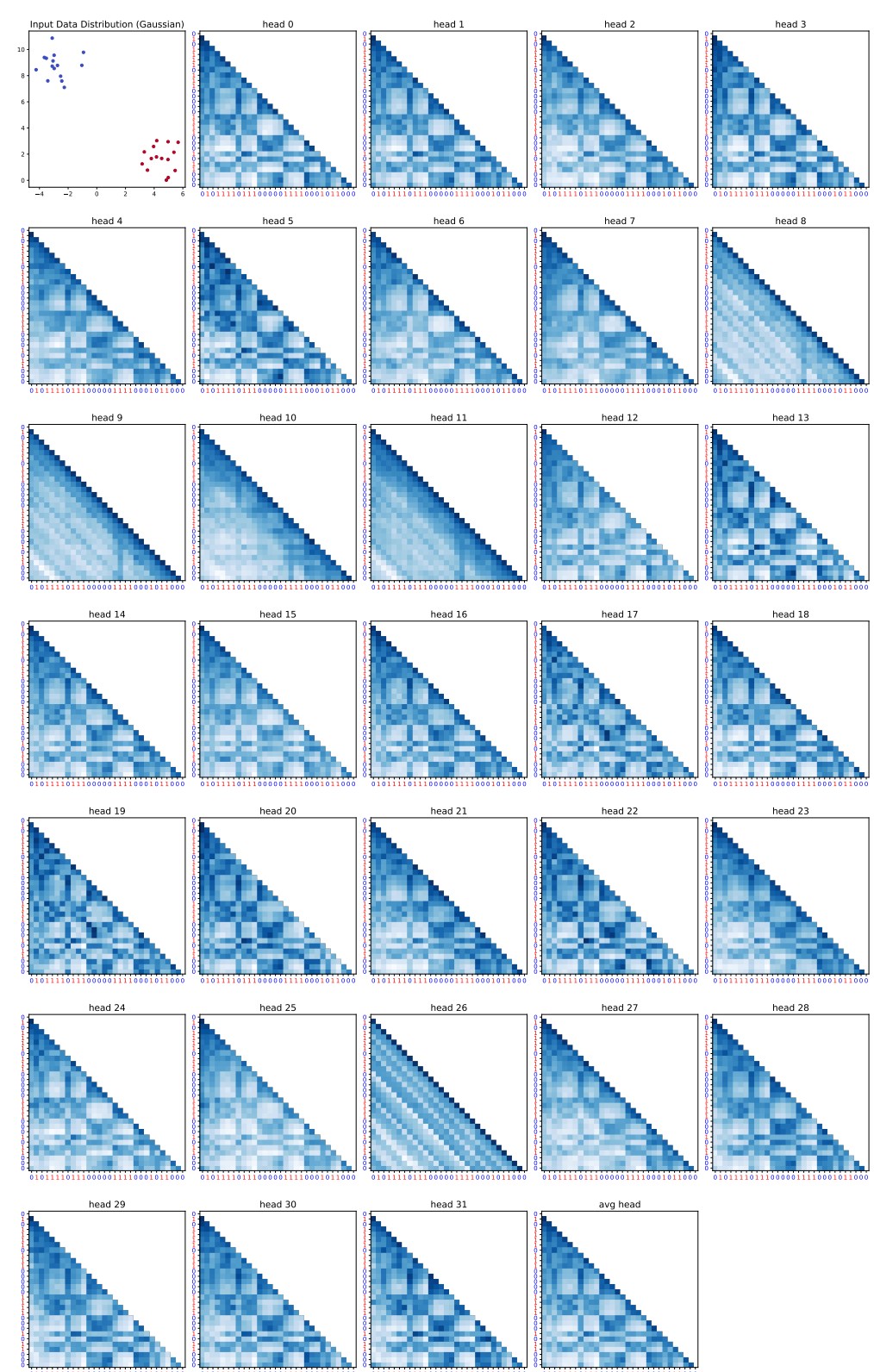

Figure 10: Attention Allocation of LLAMA-3.1-8B-INSTRUCT across attention heads at layer 15. The attention scores are logarithmized for better visualization. Each cluster is generated from a Gaussian distribution, as shown in top left.

## B.2 SPECTRAL CLUSTERING

As described in Section 3.2, we perform spectral clustering using the input-input attention score matrix $A^{II}$. We first standardize $A^{II}$ so that each row sums to one. Due to causality, early tokens cannot attend to later tokens, making the attention scores scale uneven across rows. For example, the second data point always allocates very high attention to the first one regardless of its semantic similarity. To mitigate this imbalance, we further rescale each row by the number of non-zero entries in the row. Finally, we symmetrize the matrix and the resulting matrix is used as the precomputed affinity matrix for spectral clustering. The complete preprocessing procedure is visualized in Figure 11. We use the `sklearn.cluster.SpectralClustering` implementation.

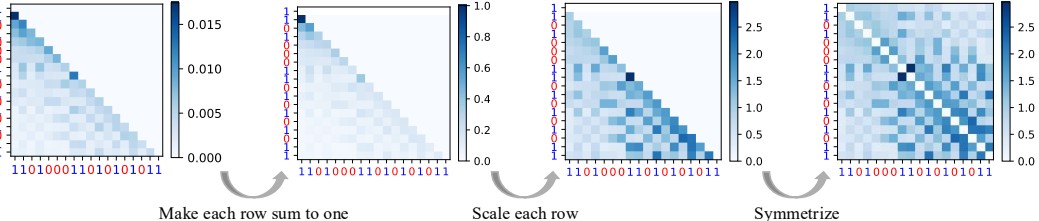

Figure 11: Preprocessing Attention Matrix for Spectral Clustering.

Table 6: Spectral Clustering using Attention Scores. Reported values are average accuracy on t-distributed test data as in Section 3, with one standard error. Models used here are pretrained LLAMA-3.1-8B-INSTRUCT and its fine-tuned checkpoint as in Section 4.1. SC represents spectral clustering using attention scores with OPT denoting the highest accuracy across all layers and L23 denoting the accuracy using a fixed layer 23 (indexing from 0). GEN represents generation using direct LLM prompting. Spectral clustering using attention achieves surprisingly competitive performance that outperforms the raw generation before finetuning.

| MODEL | METHOD | DF=1 | DF=1.25 | DF=1.5 | DF=1.75 | DF=2 | DF=5 | DF=100 |
|---|---|---|---|---|---|---|---|---|
| *num of clusters = 2, dim = 1* | | | | | | | | |
| | SC(OPT) | $0.68_{\pm0.01}$ | $0.70_{\pm0.01}$ | $0.73_{\pm0.01}$ | $0.73_{\pm0.02}$ | $0.71_{\pm0.01}$ | $0.79_{\pm0.02}$ | $0.79_{\pm0.02}$ |
| PRETRAINED | SC(L23) | $0.68_{\pm0.01}$ | $0.68_{\pm0.01}$ | $0.72_{\pm0.01}$ | $0.73_{\pm0.02}$ | $0.71_{\pm0.02}$ | $0.79_{\pm0.02}$ | $0.79_{\pm0.02}$ |
| | GEN | $0.69_{\pm0.01}$ | $0.69_{\pm0.01}$ | $0.72_{\pm0.01}$ | $0.70_{\pm0.01}$ | $0.72_{\pm0.01}$ | $0.74_{\pm0.02}$ | $0.77_{\pm0.01}$ |
| | SC(OPT) | $0.70_{\pm0.01}$ | $0.72_{\pm0.01}$ | $0.73_{\pm0.01}$ | $0.74_{\pm0.02}$ | $0.74_{\pm0.02}$ | $0.79_{\pm0.02}$ | $0.79_{\pm0.02}$ |
| FINETUNED | SC(L23) | $0.67_{\pm0.01}$ | $0.70_{\pm0.02}$ | $0.72_{\pm0.02}$ | $0.72_{\pm0.02}$ | $0.72_{\pm0.02}$ | $0.76_{\pm0.02}$ | $0.75_{\pm0.02}$ |
| | GEN | $0.85_{\pm0.01}$ | $0.86_{\pm0.01}$ | $0.87_{\pm0.01}$ | $0.89_{\pm0.01}$ | $0.90_{\pm0.01}$ | $0.91_{\pm0.01}$ | $0.94_{\pm0.01}$ |
| *num of clusters = 2, dim = 2* | | | | | | | | |
| | SC(OPT) | $0.75_{\pm0.01}$ | $0.76_{\pm0.02}$ | $0.79_{\pm0.02}$ | $0.78_{\pm0.02}$ | $0.81_{\pm0.02}$ | $0.82_{\pm0.02}$ | $0.88_{\pm0.02}$ |
| PRETRAINED | SC(L23) | $0.71_{\pm0.01}$ | $0.74_{\pm0.02}$ | $0.73_{\pm0.02}$ | $0.76_{\pm0.02}$ | $0.78_{\pm0.02}$ | $0.80_{\pm0.02}$ | $0.87_{\pm0.02}$ |
| | GEN | $0.69_{\pm0.01}$ | $0.68_{\pm0.01}$ | $0.69_{\pm0.01}$ | $0.71_{\pm0.01}$ | $0.69_{\pm0.01}$ | $0.74_{\pm0.02}$ | $0.75_{\pm0.01}$ |
| | SC(OPT) | $0.84_{\pm0.01}$ | $0.84_{\pm0.02}$ | $0.85_{\pm0.02}$ | $0.87_{\pm0.01}$ | $0.87_{\pm0.01}$ | $0.89_{\pm0.02}$ | $0.96_{\pm0.01}$ |
| FINETUNED | SC(L23) | $0.77_{\pm0.02}$ | $0.81_{\pm0.02}$ | $0.80_{\pm0.02}$ | $0.82_{\pm0.02}$ | $0.83_{\pm0.02}$ | $0.87_{\pm0.02}$ | $0.94_{\pm0.01}$ |
| | GEN | $0.92_{\pm0.01}$ | $0.94_{\pm0.01}$ | $0.93_{\pm0.01}$ | $0.95_{\pm0.01}$ | $0.94_{\pm0.01}$ | $0.96_{\pm0.01}$ | $0.98_{\pm0.01}$ |
| *num of clusters = 2, dim = 3* | | | | | | | | |
| | SC(OPT) | $0.77_{\pm0.02}$ | $0.79_{\pm0.02}$ | $0.78_{\pm0.02}$ | $0.80_{\pm0.02}$ | $0.83_{\pm0.02}$ | $0.85_{\pm0.02}$ | $0.88_{\pm0.02}$ |
| PRETRAINED | SC(L23) | $0.68_{\pm0.01}$ | $0.71_{\pm0.02}$ | $0.73_{\pm0.02}$ | $0.74_{\pm0.02}$ | $0.76_{\pm0.02}$ | $0.81_{\pm0.02}$ | $0.85_{\pm0.02}$ |
| | GEN | $0.64_{\pm0.01}$ | $0.65_{\pm0.01}$ | $0.66_{\pm0.01}$ | $0.67_{\pm0.01}$ | $0.69_{\pm0.01}$ | $0.70_{\pm0.02}$ | $0.71_{\pm0.02}$ |
| | SC(OPT) | $0.90_{\pm0.01}$ | $0.91_{\pm0.01}$ | $0.93_{\pm0.01}$ | $0.91_{\pm0.01}$ | $0.93_{\pm0.01}$ | $0.96_{\pm0.01}$ | $0.99_{\pm0.00}$ |
| FINETUNED | SC(L23) | $0.83_{\pm0.02}$ | $0.86_{\pm0.02}$ | $0.89_{\pm0.02}$ | $0.87_{\pm0.02}$ | $0.91_{\pm0.01}$ | $0.95_{\pm0.01}$ | $0.97_{\pm0.01}$ |
| | GEN | $0.96_{\pm0.01}$ | $0.97_{\pm0.01}$ | $0.98_{\pm0.00}$ | $0.96_{\pm0.01}$ | $0.98_{\pm0.00}$ | $0.99_{\pm0.00}$ | $1.00_{\pm0.00}$ |

## C ADDITIONAL EXPERIMENT DETAILS AND RESULTS OF IMAGE CLUSTERING

### C.1 POOLING

Table 7: Pooling kernel size and corresponding per-image token length. The original pixel size is 384x384 with a patch size of 14, resulting in 27x27(729) image tokens.

|  | POOLING KERNEL | TOKEN LENGTH |
| --- | --- | --- |
| DEFAULT | 1x1 | 27 x 27 (729) |
| LARGE | 2x2 | 13 x 13 (169) |
| MEDIUM | 3x3 | 9 x 9 (81) |
| SMALL | 9x9 | 3 x 3 (9) |

### C.2 OUT-OF-DOMAIN IMAGE DATASETS

To test the generalization capability of the model, we include two more image datasets from Cross-Domain Few-Shot Learning (CD-FSL) Benchmark Guo et al. (2020).

- Plant Disease Mohanty et al. (2016): Leaves of different trees that are healthy or have different crop diseases. We construct 100 clustering samples based on the plant names, where each sample contains 10-30 images from 3 random classes.
- EuroSAT Helber et al. (2019): Satellite images of different land use and land cover classes. We construct 100 clustering samples where each sample contains 10-30 images from 3 random classes.

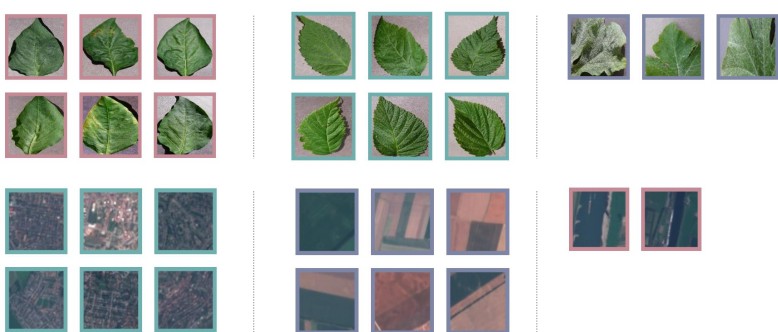

Figure 12: Example of Plant Disease and EuroSAT datasets. The color of frame represents different clusters predicted by our model. Our model can generalize to these images that are quite different from ImageNet.

### C.3 ATTENTION

Similar as the numeric experiments in Section 3.2, we visualize the attention allocation for image clustering below (Figure 13). The model used here is fine-tuned model (medium) as in Section 4.2. The attention scores have block structures that roughly align with the ground-truth identities in intermediate layers. We notice that the allocation of attention weights can be uneven within one cluster, where representative samples are assigned with higher weights. The attention patterns for images are generally more complicated than those for synthetic low-dimensional data due to the semantically rich information in images.

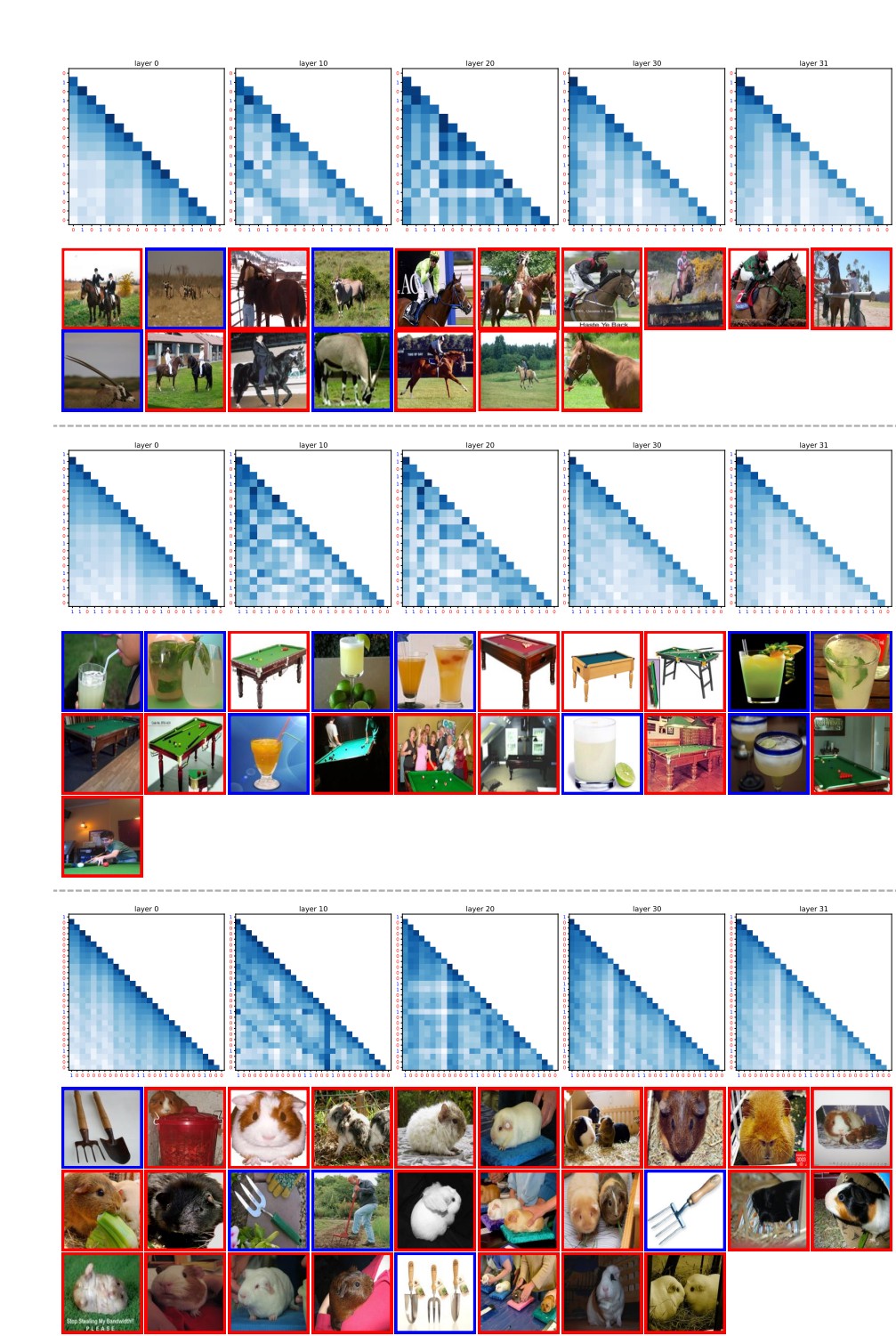

Figure 13: Attention Allocation of Image Clustering. Different colors represent different clusters.

# D  ADDITIONAL RESULTS FOR CONDITIONAL IMAGE CLUSTERING

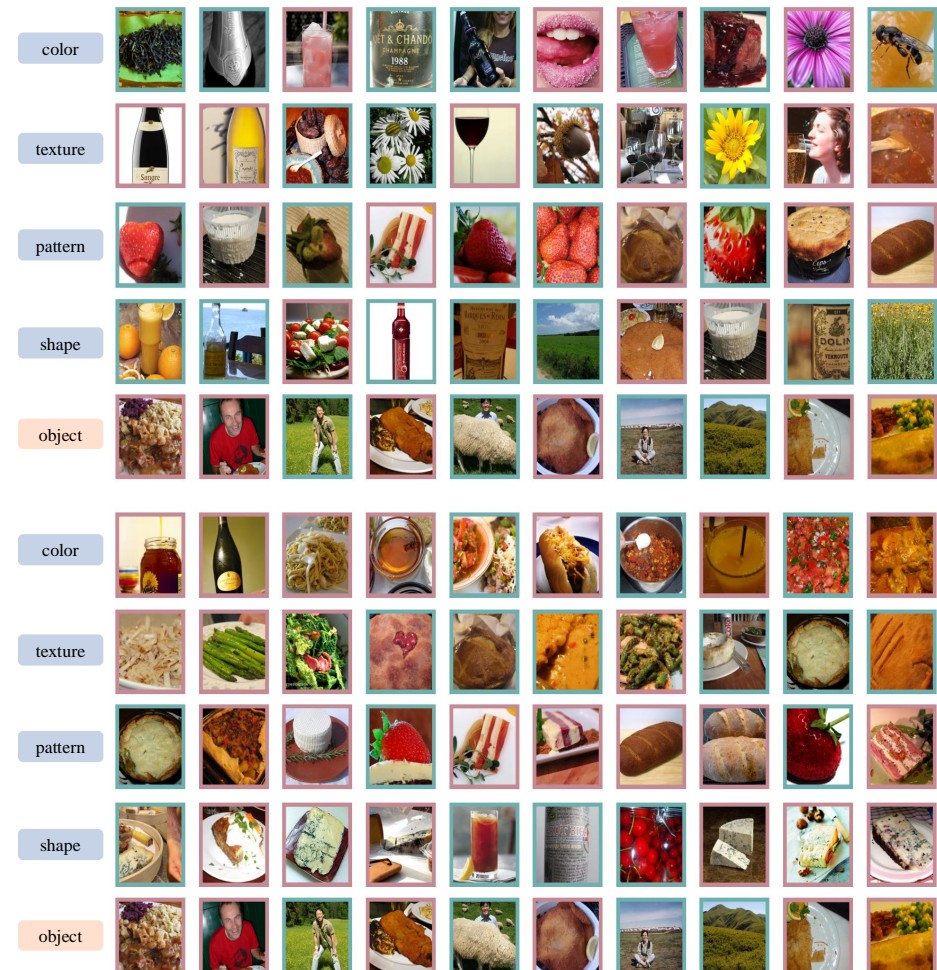

Figure 14: Examples of ICC on ImageNet-with-Attributes. The color of the frame indicates different clusters predicted by our model. Most of the images contain multiple objects, making the task more challenging.

# E    ADDITIONAL EXPERIMENTS DETAILS

## E.1    NUMERIC CLUSTERING

For all Llama AI@Meta (2024) and Qwen Bai et al. (2023) models, we use the implementation and checkpoints from HuggingFace. Specifically, we test the following models in Section 3.1.

- `meta-llama/Llama-3.2-1B`,    `meta-llama/Llama-3.2-1B-Instruct`, `meta-llama/Llama-3.2-3B`, `meta-llama/Llama-3.2-3B-Instruct`,    `meta-llama/Llama-3.1-8B`, `meta-llama/Llama-3.1-8B-Instruct`;

- `Qwen/Qwen2.5-7B-Instruct`,    `Qwen/Qwen2.5-3B-Instruct`, `Qwen/Qwen2.5-14B-Instruct`

We then fine-tune Llama models in Section 4.1. We use LoRA Hu et al. (2021) with $r = 64$ and $alpha = 16$ for finetuning. We use an initial learning rate of $lr = 5e - 4$ with a cosine learning rate scheduler. We use one A100 and a batch size of 32. We save the checkpoint with the lowest validation loss.

## E.2    IMAGE CLUSTERING

For both unconditional (Section 4.2) and conditional image clustering (Section 5), we use `llava-interleave-qwen-7b-hf` as our base model and use the implementation and checkpoints from HuggingFace. We use LoRA Hu et al. (2021) with $r = 64$ and $alpha = 16$. We use an initial learning rate of $lr = 5e - 4$ with cosine learning rate scheduler. We use two A100s and an effective total batch size of 32. For models with smaller pooling kernels (and thus higher per-image token length), we use gradient accumulation (Table 8). Our finetuning code is adapted from Zhang & Lin. We save the checkpoint with the lowest validation loss.

Table 8: Pooling Kernel Size and Per-Device Batch Size.

|  | POOLING KERNEL | TOKEN LENGTH | PER-DEVICE BATCH SIZE | GRADIENT ACCUMULATION |
|---|---|---|---|---|
| LARGE | 2x2 | 13 x 13 (169) | 4 | 4 |
| MEDIUM | 3x3 | 9 x 9 (81) | 8 | 2 |
| SMALL | 9x9 | 3 x 3 (9) | 8 | 2 |

# F    THE USE OF LARGE LANGUAGE MODELS (LLMS)

We only use an LLM for proofreading. We have carefully reviewed all suggestions and take full responsibility for the final content of the paper.

