# OpenReview forum: "In-Context Clustering with Large Language Models"
_ICLR.cc/2026/Conference — Submitted to ICLR 2026_

### Official Review · Reviewer_BxBQ · 2025-10-16

**Soundness:** 1
**Presentation:** 3
**Contribution:** 2
**Rating:** 2
**Confidence:** 5

**Summary:**

This manuscript proposes In-Context Clustering (ICC), an LLM-based method that performs clustering without predefined similarity functions. ICC uses the attention mechanism of pretrained LLMs to capture context-dependent relationships among inputs across modalities. The authors show that LLMs or multimodal LLMs exhibit zero-shot clustering ability in numercal and visual data. With additional fine-tuning using LoRA with a Next Token Prediction loss, the experiments showed that ICC achieved improved performance on both numeric and image datasets. Moreover, ICC supports text-conditioned image clustering that allows prompt-based control on the clustering process.

**Strengths:**

Major Strengths:
- The writing and organization of this manuscript is clear and easy to follow (yet, it's better to add necessary details to make the main paper self-contained; jumping to figures in appendix from main paper is not very enjoable)

- The experiment that visualizes the attention allocation of input data and generated cluster labels at an intermediate layer is very interesting and innovative. This will give a good support to the clustering mechanism behind LLMs when in-context clustering is used.

- The presentation and visualization of this manuscript is clear and visually enjoyable.

**Weaknesses:**

Major Weakness:

- **My primary concern about this manuscript lies in the validity of its central claim** that “in-context clustering with large language models (LLMs) performs as well as or better than” traditional clustering methods such as K-means, spectral clustering, or DBSCAN, or other related methods. The rationale is as follows: clustering algorithms are designed to handle and explore unlabeled, unseen, and novel data—such as new concepts, observations, or protein structures—across diverse modalities. In contrast, the proposed “in-context clustering with LLMs” method fundamentally depends on pre-trained LLMs or multimodal LLMs and, consequently, on the massive datasets and implicit clustering criteria, and of course, clustering centroids, these models have already encountered during training. Therefore, it is unclear how well the proposed method would perform when the data is genuinely novel, unseen or out-of-distribution. This data coverage limitation also raises concerns about the validity of the claimed “zero-shot” setting in the experiments.  In contrast, traditional clustering methods such as kmeans can be easily adapted to truly novel and unseen data.

- **The reviewer found the following statement to be an overclaim:** At Line 73, the authors state: “We believe that this ability to change the way clustering is done based on different prompts makes ICC, and this research direction, particularly compelling.”  **In fact, text-conditioned or prompt-steered clustering using LLMs paradigm was first proposed by IC|TC [1], and subsequently explored in [2,3,4]. Moreover, [5] further extended this line of work by enabling automatic discovery of clustering conditions from data using LLMs.** So, such innovation and capability has already been proposed and studied by the community recently. **The authors should appropriately acknowledge prior research contributions rather than implying that this innovation originates solely from their proposed ICC method.**

- Several highly relevant studies, including [3, 4, 5], are missing from the literature review and discussion.

- Compared to IC|TC [1] and [2], the novelty of the proposed ICC method is quite limited, as it mainly adds an additional fine-tuning component.

- **Regarding the “Zero-shot In-context Clustering” experiments in Section 3.1: are they truly zero-shot? The prompt template (Lines 144–146) explicitly provides the number of clusters to the model.** This information constitutes a strong prior about the data structure, meaning the model already **knows how many ground-truth groups exist in the dataset**. With such prior knowledge given, the zero-shot nature of the setting is questionable. In real-world zero-shot scenarios, the number of clusters is often *unknown*.

- **The experimental setup and baseline comparisons in Section 4.2 (Table 2) and Section 5 (Table 3) appear to be unfair.** The IC|TC baseline is training-free and uses LLMs directly without fine-tuning (e.g., GPT-3.5-turbo). In contrast, the proposed ICC method  either (1) uses GPT-4o, which is a much stronger model, or (2) llava-interleave-qwen-7b-hf includes further fine-tuning on data drawn from a similar distribution. Comparing the GPT-4o model and fine-tuned llava-interleave-qwen-7b-hf to a training-free GPT-3.5-turbo baseline is not fair, as it conflates improvements due to model scale and additional tuning. Consequently, the conclusions drawn from this comparison are not well supported.

- Further, the reviewer questions how ICC would compare against traditional clustering methods using strong vision features. **For example, what is the clustering performance of K-means when using features extracted from DINOv3-ViT-7B/16?** Would ICC—relying on a significantly larger model—still outperform DINOv3-based clustering under comparable settings?


[1] Kwon, Sehyun, et al. "Image clustering conditioned on text criteria." In ICLR, 2024.

[2] Luo, Yulin, et al. "Llm as dataset analyst: Subpopulation structure discovery with large language model." In ECCV, 2024.

[3] Yao, Jiawei, Qi Qian, and Juhua Hu. "Customized multiple clustering via multi-modal subspace proxy learning." In NeurIPS, 2024.

[4] Yao, Jiawei, Qi Qian, and Juhua Hu. "Multi-modal proxy learning towards personalized visual multiple clustering." In CVPR, 2024.

[5] Liu, Mingxuan, et al. "Organizing unstructured image collections using natural language." Arxiv preprint, 2024.

**Questions:**

Minor questions are described in the following:
- The authors claim that prior similarity-based clustering methods cannot capture “context.” However, no proof, reference, or experimental evidence is provided to support this claim in either the textual or visual modality. In fact, many earlier methods in text clustering, including classical probabilistic models such as LDA [6], explicitly aim to model contextual information to group documents by topic. Similarly, in the vision domain, when images are represented through learned or encoded features, it is unclear to the reviewer why such representations would be inherently incapable of capturing context.

- Regarding the evaluation metrics in Section 3.1: while it is standard practice to compute clustering accuracy using the Hungarian linear assignment, this metric can be easily biased due to its matching paradigm. Since LLMs are capable of producing textual labels for each cluster, the authors could consider an alternative approach that approximates classification accuracy for a more direct comparison.

- At Line 189, the authors state: “We also observe that instruction tuning improves the overall accuracy.” However, the dataset used for instruction tuning, and the details of how this tuning was performed, are not specified in the paper. Without this information, the result cannot be properly interpreted or reproduced.

- Please explain what is “df” (degree of freedom) in Section 3.1. It is not explained in the manuscript.

[6] Blei, David M., Andrew Y. Ng, and Michael I. Jordan. "Latent dirichlet allocation." Journal of machine Learning research 3.Jan (2003): 993-1022.

---

> ### Author Response · Authors · 2025-12-03
>
> We thank Reviewer BxBQ for the review. We believe the reviewer's score is based on several fundamental misunderstandings about our method's architecture, our experimental setup, and our results. We clarify these major points first.
>
> ### Missing literature and novelty
>
> We thank the reviewer for pointing out these related works [3,4,5]. We will add them to our related work section. However, we must clarify that these methods do not diminish our novelty and some of them are not comparable.
>
> - [1, 2, 5] are all caption-based methods. As we have emphasized in our paper (L105-124, L362-364, L374-377, L452-454), this reliance on an intermediate text caption is a major bottleneck, which our method explicitly avoids.
> - [3,4]  are not directly comparable for two main reasons: (i) [3,4] rely on GPT-4 to generate a set of "reference words" (i.e., candidate cluster labels) for a given user-defined concept (e.g., "common colors of fruit" or "car types" ). This approach inherently limits their applicability to domains where such an enumeration of categories is feasible and semantically meaningful. This is reflected in their use of simple, specific datasets like fruits, cars, and flowers. In contrast, our method operates under the more general assumption of an unknown data domain. (ii) [3,4] involve an optimization process during clustering, which makes inference time slow. Specifically, [4] requires 1000 epochs of training and [3] employs a two-phase iterative approach where the first proxy learning phase takes 100 epochs and the second clustering phase takes 10 epochs. In contrast, **ICC does not involve an additional optimization loop during the clustering step itself.**
>
> > My primary concern about this manuscript lies in the validity of its central claim that “in-context clustering with large language models (LLMs) performs as well as or better than” traditional clustering methods [...]
>
> We respectfully disagree with this concern. LLMs show great generalizability due to the massive data seen during pretraining. Our method leverages pretrained LLMs, which have learned a flexible, dynamic, and semantic similarity mechanism via attention. This learned mechanism is more adaptable to novel data drawn from a random distribution than a rigid prior used in classical methods.
>
> > Regarding the “Zero-shot In-context Clustering” experiments in Section 3.1: are they truly zero-shot? The prompt template (Lines 144–146) explicitly provides the number of clusters to the model.
>
> **We use "zero-shot" in the standard ICL sense: zero in-context examples (i.e., no (data, label) pairs are provided in the prompt).** Providing k (the number of clusters) is a standard hyperparameter for most clustering algorithms (e.g., k-means, Spectral Clustering, GMM) and does not violate this definition.
>
> > The experimental setup and baseline comparisons in Section 4.2 (Table 2) and Section 5 (Table 3) appear to be unfair.
>
> The models are different because the tasks are different. We must use a multimodal model like gpt-4o for our zero-shot ICC baseline because our method directly takes images as input. The core limitation of IC|TC and all caption-based methods, which we explicitly challenge, is the captioning bottleneck. Inevitable and irrecoverable information loss occurs when a complex image is reduced to a short text caption. Using a stronger LLM for the text-clustering part of IC|TC would not compensate for the rich visual information that was already lost during captioning.
>
> > The authors claim that prior similarity-based clustering methods cannot capture “context.”
>
> We focus on the flexibility of LLMs that can dynamically adapt its clustering logic at inference time based on the specific inputs and instructions in the context. While LDA captures "context" as latent topics within a fixed corpus, it cannot dynamically adapt its clustering logic based on a natural language instruction (e.g., changing from "cluster by color" to "cluster by shape") without retraining.
>
> > dataset used for instruction tuning, and the details of how this tuning was performed, are not specified in the paper.
>
> The "instruction-tuning" refers to the instruction-tuned models, as specified in fig7 caption. Instruction-tuning here refers to the standard training where a base LLM is finetuned on a dataset of (instruction, response) pairs to learn to follow instructions. We use instruction-tuned models released by the Llama team. Although the Llama team doesn’t disclose exact instruction data and training details, we list the HuggingFace model IDs for all models used in Appendix E.1 for reproducibility.
>
> > Please explain what is “df” (degree of freedom) in Section 3.1. It is not explained in the manuscript.
>
> df is used in the context of the t-distribution (e.g. L147-153, figure 2 caption), a parameter of a t-distribution that controls the heavy-tailedness.

---

### Official Review · Reviewer_UQsK · 2025-10-31

**Soundness:** 2
**Presentation:** 3
**Contribution:** 1
**Rating:** 2
**Confidence:** 4

**Summary:**

This paper introduces In-Context Clustering (ICC), a novel method that leverages LLMs for clustering of images and numerical data. The authors show that the LLM’s attention mechanism captures complex relationships between inputs that can be used for clustering with spectral clustering. Further improvements are obtained through fine-tuning with next-token prediction, extending the method to numeric and image data. Additionally, text-conditioned image clustering is demonstrated where multiple different clusterings can be extracted based on the design of the prompt.

**Strengths:**

- The paper addresses a highly relevant research area, presenting an approach that enables more user-guided clustering through prompt-based interactions with LLMs
- I found it interesting that the method works well with numerical data as input for clustering via LLMs, although this capability appears to be constrained to lower-dimensional data
- Employing the attention matrix derived from the LLM as input for spectral clustering is an interesting insight
- Experiments show benefits across different datasets and modalities

**Weaknesses:**

## Soundness

The authors limit their comparison to a single classical clustering algorithm, namely K-Means, which serves already as a strong baseline in Table 2 and 3. Based on this I am missing the comparison to different classical algorithms like Expectation-Maximization Clustering, DBSCAN or its popular extension HDBSCAN to see if "simpler" baselines can outperform the proposed method.


## Novelty

My main concern with this paper lies in its limited novelty. Several recent works have already explored closely related ideas, particularly the use of prompting and multimodal representations to induce or control clustering behavior, but the authors only compare to IC|TC. For example, prior studies such as

- **Jiawei et al. "Multi-modal proxy learning towards personalized visual multiple clustering." CVPR 2024.**

- **Jiawei et al. "Customized multiple clustering via multi-modal subspace proxy learning." NeurIPS (2024): 82705-82725.**

- **Stephan et al. Text-Guided Image Clustering. EACL (1) 2024: 2960-2976**

- **Stephan et al (2024). Text-Guided Alternative Image Clustering. In Proceedings of the 9th Workshop on Representation Learning for NLP (RepL4NLP-2024) (pp. 177-190).**

already use prompt-based approaches to obtain one or multiple clusterings conditioned on different attributes or textual guidance.

Moreover, recent works such as

- **Gadetsky et al: Large (Vision) Language Models are Unsupervised In-Context Learners. ICLR 2025**

- **Gadetsky et al: Let Go of Your Labels with Unsupervised Transfer. ICML 2024**

demonstrate already that large (vision) language models can perform unsupervised in-context learning and clustering without explicit supervision.

Taken together, these prior works already explore the use of LLMs and in-context mechanisms for unsupervised or text-guided clustering, which significantly overlaps with the proposed In-Context Clustering (ICC) framework. The authors should therefore clearly differentiate their method from these existing methods and compare to them in benchmarking experiments. Further, a dedicated discussion of what is conceptually and technically novel about ICC compared to these earlier contributions is missing.

**Questions:**

- In what key methodological ways does your approach differ from the prior works referenced in the weaknesses section? What are the key contributions of your method?
- The current comparison is limited to k-Means and IC|TC. How does your algorithm perform relative to other recently proposed methods mentioned above?
- How do other classical clustering algorithms compare to your approach? Are there scenarios in which simpler baselines outperform ICC, and if so, under what circumstances? More broadly, when might traditional methods be sufficient compared to LLM-guided clustering?

---

> ### Author Response · Authors · 2025-12-03
>
> We thank Reviewer UQsK for the feedback. We appreciate the opportunity to clarify the difference between our work and prior works and its performance relative to classical baselines.
>
> ## Soundness: Comparison with more classical algorithms
>
> Thanks for the suggestion! We added HDBSCAN, Spectral, and GMM results and found ICC consistently outperforms them on ImageNet. Classical clustering algorithms can be competitive when the underlying feature space is low-dimensional, well-separated, and nearly Euclidean. For example, when clusters are approximately spherical Gaussian(favoring k-means) or exhibit strong density gaps (favoring DBSCAN), these methods may achieve performance similar to or occasionally better than ICC. ICC wins in few-shot scenarios involving semantically rich, naturalistic data, complementing classical methods optimized for structured large-scale datasets.
>
> | ImageNet | c=2 | c=3 | c=4 | c=5 |
> |---|---|---|---|---|
> | HDBSCAN | 84.46 | 80.85 | 81.87 | 78.28 |
> | Spectral | 61.90 | 49.24 | 44.08 | 42.49 |
> | GMM | 92.67 | 84.72 | 82.48 | 80.39 |
> | k-means | 89.43 | 82.09 | 79.07 | 77.96 |
> | ICC(medium) | **98.26** | **95.92** | **91.62** | **84.92** |
>
> ## Novelty: Comparison with prior works
>
> We thank the reviewer for highlighting these relevant works. We agree that comparing our work against these prior works strengthens the paper. While some of prior works focus on LLM-based image clustering, we emphasize that **our contribution is not limited to an image clustering application but rather investigates the fundamental capability of LLMs to cluster data from diverse distributions in-context**. We clarify the fundamental conceptual distinctions between our work and the referenced papers below, and will include them in our related work.
>
> 1. Distinction from prompt-guided visual clustering (Jiawei et al.)
>
> While Jiawei et al. [1,2] utilize prompts to guide clustering, our method differs in generalizability and inference efficiency:
> - [1,2] rely on GPT-4 to generate a set of "reference words" (i.e., candidate cluster labels) for a given user-defined concept (e.g., "common colors of fruit" or "car types" ). This approach inherently limits their applicability to domains where such an enumeration of categories is feasible and semantically meaningful. This is reflected in their use of simple, specific datasets like fruits, cars, and flowers. In contrast, our method operates under the more general assumption of an unknown data domain.
> - [1,2] involve an optimization process during clustering, which makes inference time slow. Specifically, [1] requires 1000 epochs of training and [2] employs a two-phase iterative approach where the first proxy learning phase takes 100 epochs and the second clustering phase takes 10 epochs. In contrast, **ICC does not involve an additional optimization loop during the clustering step itself**.
>
> 2. Distinction from caption-based image clustering  (Stephan et al.)
>
> Stephan et al. [3,4] generate image captions and then use k-means to cluster the embeddings of these captions. The idea is similar to IC|TC and SSD-LLM, which generate image captions and then use LLMs to cluster these captions. As we emphasized in our paper, these caption-based methods suffer from inevitable information loss. **ICC bypasses the captioning bottleneck**, allowing the model to cluster based on rich visual semantics rather than intermediate textual descriptions.
>
> 3. Distinction from unsupervised adaptation methods (Gadetsky et al.)
>
> Although Gadetsky et al. [5,6] also explore unsupervised learning, their objectives and mechanics are fundamentally different from ICC.
> - [5] is a joint inference framework where the model reasons over multiple inputs to maximize their joint probability. One of their key contribution is unsupervised ICL, which iteratively refines predictions by using the model’s own outputs from previous iterations, wheras traditional ICL uses ground truth labels for in-context examples. Note that both [5] and traditional ICL focus on supervised tasks (e.g. classification). Our in-context clustering is fundamentally different from [5] as it aims to use ICL to solve an unsupervised learning task (i.e. clustering), and doesn’t require any training/refinement during inference.
> - [6] is a completely different work that doesn’t use ICL or even LLMs at all. It operates on fixed feature embeddings to search for labelings that maximize the margin of linear classifiers within the representation space.
>
> |  | task | GT labels | Inference-time |
> |---|---|---|---|
> | (traditional) ICL | Classification (supervised)  | Required in context | One forward pass |
> | Unsupervised ICL [5]  | Classification (supervised)   | Not required (uses self-generation) | Iterative refinement  |
> | In-context Clustering | Clustering(unsupervised) | Not required | One forward pass |

---

### Official Review · Reviewer_9K37 · 2025-11-01

**Soundness:** 2
**Presentation:** 2
**Contribution:** 2
**Rating:** 4
**Confidence:** 3

**Summary:**

The article introduces In-Context Clustering (ICC), which extends the in-context learning paradigm to unsupervised clustering tasks. The authors demonstrate that large language models (LLMs) can perform zero-shot clustering on text-encoded numeric data and images by leveraging their attention mechanisms to capture complex relationships between inputs. The work introduces fine-tuning strategies using next token prediction (NTP) loss to enhance clustering performance, particularly for heavy-tailed distributions and semantically rich data. Additionally, ICC enables text-conditioned image clustering, allowing users to specify clustering criteria through natural language prompts.

**Strengths:**

- The paper is clear about its motivation with sufficient significance and quality.
- The paper makes a compelling case for extending in-context learning to unsupervised settings. The ability to perform clustering without predefined similarity measures through prompting is innovative and addresses real limitations of classical methods.
- The zero-shot clustering results on t-distributed data with varying degrees of freedom convincingly demonstrate that LLMs can outperform k-means when Gaussian assumptions are violated. The performance gains are particularly striking for heavy-tailed distributions.
- The visualization and analysis of attention matrices revealing emergent cluster structures (Section 3.2) provides valuable insights into the internal mechanisms. The finding that spectral clustering on attention matrices achieves 85% accuracy before fine-tuning while generation only reaches 74% is particularly intriguing.

**Weaknesses:**

- The paper insufficiently addresses the computational limitations for practical deployment. With O(n²) attention complexity and token limits, how does ICC handle datasets with thousands of points? The average pooling for images seems like a band-aid solution that could lose critical fine-grained information.
- While the empirical results are good, the paper lacks theoretical analysis of when and why ICC works. What properties of the attention mechanism enable clustering? Under what conditions might ICC fail?
- This is the most critical weakness of the paper. For image clustering, the comparison is limited to k-means and IC|TC. Missing comparisons with modern deep clustering methods (e.g., SCAN, NNM, SwAV, or other self-supervised approaches) makes it difficult to assess the true performance gains.
- The fine-tuning data generation process using t-distributions with random parameters seems arbitrary. How sensitive is performance to this choice?
- No ablation studies on key design choices (e.g., impact of different pooling strategies, prompt variations)
- Figure quality could be improved - some attention visualizations are difficult to interpret
- The related work section could better position this work relative to recent advances in foundation models for clustering

**Questions:**

- How does performance degrade as the number of data points approaches context limits? Have you experimented with hierarchical clustering or other strategies to handle larger datasets?
- How robust is ICC to prompt variations? The paper uses a simple template "Cluster the following data into {k} clusters" - have you tested more sophisticated prompting strategies or chain-of-thought reasoning?
- Can you provide any theoretical justification for why attention patterns correspond to cluster structure? Is there a connection to graph-based clustering methods or spectral theory?
- What types of clustering problems does ICC struggle with? For instance, how does it handle clusters with varying densities, non-convex shapes, or hierarchical structures?

---

> ### Author Response · Authors · 2025-12-03
>
> We thank Reviewer 9K37 for the review. We are grateful for the positive feedback acknowledging **our paper's clear motivation, significance and innovation of ICC**. We will address each of the listed weaknesses and questions below
>
> > The paper insufficiently addresses the computational limitations for practical deployment. With O(n²) attention complexity and token limits, how does ICC handle datasets with thousands of points?
>
> We acknowledge that scalability is a current limitation (L477-480), but propose ICC as a novel approach that excels where classical methods may not. ICC is particularly useful in low-shot, semantically rich clustering tasks where flexibility and contextual understanding are more critical than processing massive datasets. It complements, rather than replaces, classical methods optimized for large-scale structured data. The challenge of O(n²) attention complexity and token limits stems from the underlying LLM architecture, not a flaw in the ICC methodology itself. We have already taken concrete steps to improve efficiency via an average pooling strategy that significantly reduces token usage for images. As we discussed in the limitation section, the rapid progress in long-context models provides a clear path to addressing this limitation in future work.
>
> > While the empirical results are good, the paper lacks theoretical analysis of when and why ICC works.
>
> We agree and acknowledge this lack of theoretical understanding in our conclusion (L481-485). This paper is one of the initial attempts that validate in-context clustering via empirical experiments and a full theoretical analysis is beyond the scope of this work. We note that prior work on LLM-based clustering(e.g. IC|TC, SSD-LLM)  also lacks this deep theoretical analysis. To gain an understanding of the internal mechanism behind ICC, we provided a novel attention analysis in Sec3.2 where we find emergent cluster structures in the attention matrices. We believe this analysis is a valuable contribution that could benefit future research on understanding and improving the attention mechanism.
>
> > No ablation studies on key design choices (e.g., impact of different pooling strategies, prompt variations)
>
> In fact, we have included ablation studies on pooling in the original submission. Tables 2 and 3 show results for "SMALL," "MEDIUM," and "LARGE" with Appendix C.1 (Table 7) explicitly details what these correspond to different pooling.

---

### Official Review · Reviewer_auZ2 · 2025-11-01

**Soundness:** 1
**Presentation:** 2
**Contribution:** 1
**Rating:** 2
**Confidence:** 4

**Summary:**

This paper proposes and tests using LLMs for “in-context clustering”, where (multimodal) LLMs are presented sequences and are tasked with assigning cluster labels (conditioned on a priori cluster count) to each element. The experiments include (1) synthetic numerical clustering, where points sampled from mixtures of low-dimensional t-distributions with varying degrees of freedom; (2) attention-based analysis, where token-level attention maps are treated as affinity matrices for spectral clustering; (3) LoRA fine-tuning, where the model is trained via next-token prediction on synthetic text prompts containing sample–label pairs; and (4) image experiments, where images and captions from ImageNet are clustered through textual prompts. All experiments report Hungarian-aligned accuracy against true labels and compare only to simple baselines like k-means.

**Strengths:**

- The experiments are reproducible and clearly presented
- Analysis of attention affinity matrices is interesting

**Weaknesses:**

- Performance gains are unsurprising: the fine-tuned models are trained on synthetic mixtures drawn from the same/very similar distribution family as the evaluation sets, so improvement simply reflects distributional overlap, not generalizable clustering ability.
- "Classical methods often rely on predefined measures" - I don't agree with this. For example, embedding models trained with contrastive learning transform data onto a low dimensional manifold where local distance meaningfully represents semantic difference.
- Minimal novelty. The pipeline and evaluation duplicate prior IC|TC work, with only superficial framing changes (“in-context” language).
- Accuracy via Hungarian matching inflates scores and hides near-chance performance. Consider using ARI/NMI as well.

**Questions:**

- Does attention-spectral remain strong under permutation of item order and different prompt formats? Show stability across layers/heads with an automatic selection rule.
- How do CLIP/DINOv2 features + spectral/DBSCAN/GMM compare under the same data, including conditional setups via text features?

---

> ### Author Response · Authors · 2025-12-03
>
> We thank Reviewer auZ2 for the feedback. We must first correct a fundamental misunderstanding in the reviewer's summary which appears to be the root cause of the main concerns about novelty.
>
> > Minimal novelty. The pipeline and evaluation duplicate prior IC|TC work
>
> The reviewer's summary states that "(4) image experiments, where images and captions from ImageNet are clustered...". This is factually incorrect and misrepresents our core contribution because **our method does not use captions at all**. This is the key difference between our work and prior work IC|TC. As shown in Figure 4 and described in detail in L301-309, we project image embeddings directly into the LLM's embedding space, and the LLM reasons directly over these embeddings. By contrast, IC|TC is a caption-based method, which converts images into captions and thus its performance is bottlenecked by the intermediate caption quality (see paper L105-124, L362-364, L374-377, L452-454).
>
> Our pipeline and evaluation are fundamentally different from IC|TC. **We study a more fundamental problem of the LLM's capability to cluster data from diverse distributions in-context**, not a "superficial framing change." Besides text-conditioned image clustering, we also include evaluation on numeric data and unconditional image clustering, a finetuning strategy to enhance clustering capability, and novel analysis of emergent attention structures.
>
> > Performance gains are unsurprising: the fine-tuned models are trained on synthetic mixtures drawn from the same/very similar distribution family as the evaluation sets, so improvement simply reflects distributional overlap, not generalizable clustering ability.
>
> We would like to point out that our experiments were **specifically designed to test generalization beyond the training distribution**.
> - Numeric Data (Table 1 & 4): We finetune on df $\in$ {1, 2, 5, 100} but test on unseen dfs {1.25, 1.5, 1.75}. Furthermore, Table 4 (Appendix) shows our model, trained only on t-distributions, generalizes well to a completely different lognormal distribution, outperforming baselines.
> - Image Data (Table 2): We finetune on a subset of ImageNet classes but test on **unseen** ImageNet classes and, crucially, on an unseen number of clusters (c=5) and on **OOD datasets: Plant Disease, EuroSAT, and Stanford 40 Actions**. The strong performance on these datasets supports the generalization of our method.
>
> > "Classical methods often rely on predefined measures" - I don't agree with this. For example, embedding models trained with contrastive learning transform data onto a low dimensional manifold where local distance meaningfully represents semantic difference.
>
> We want to clarify that the inputs to classical methods are embeddings from a strong pretrained visual encoder, **SigLip**, in our image experiments (L358-359). Even when a strong, learned embedding is used, k-means yields unsatisfying performance due to its predefined similarity measures (i.e., Euclidean distance). This is because the learned embedding spaces are not guaranteed to be spherical Gaussian, which is precisely the scenario our numeric experiments are designed to test. As Figure 2 shows, on heavy-tailed t-distributions (a non-Gaussian case), k-means fails while our method excels. In the image experiment, we apply k-means to the exact same SigLip features that our ICC model uses. The results in Table 2 demonstrate that our method is superior even when the baseline uses the same high-quality features. Furthermore, the "learned embedding + k-means" pipeline is fixed to one clustering result. One advantage of our method is its flexibility as the clustering criterion can be changed dynamically at inference time via a text prompt (e.g., "cluster by color" vs. "cluster by class"), a capability classical methods lack.
>
> > How do CLIP/DINOv2 features + spectral/DBSCAN/GMM compare under the same data, including conditional setups via text features?
>
> Thanks for your suggestion. In the experiments in our paper, we use SigLip, a CLIP-like image–text model that replaces CLIP’s softmax contrastive loss with a sigmoid loss to improve training stability and alignment [1]. We choose it because it has outperformed CLIP and is widely used as visual encoder. As requested, we added experiments on spectral, HDBSCAN(as it gives better performance than DBSCAN) and GMM, and showed that our ICC still outperforms these popular strong classical methods.
>
> | ImageNet  | c=2 | c=3 | c=4 | c=5 |
> |---|---|---|---|---|
> | HDBSCAN | 84.46 | 80.85 | 81.87 | 78.28 |
> | Spectral | 61.90 | 49.24 | 44.08 | 42.49 |
> | GMM | 92.67 | 84.72 | 82.48 | 80.39 |
> | k-means | 89.43 | 82.09 | 79.07 | 77.96 |
> | ICC(medium) | **98.26** | **95.92** | **91.62** | **84.92** |
>
> [1] Sigmoid Loss for Language Image Pre-Training (Zhai et al.)

---

### Author Response · Authors · 2025-12-03

We sincerely thank all reviewers for their time and valuable feedback on our work. We are grateful for the positive comments, including the **paper's clarity (9K37, BxBQ), the novelty and significance of extending ICL to an unsupervised setting (9K37), and insightful analysis of the LLM's attention mechanism (auZ2, 9K37, UQsK, BxBQ)**.

The most critical point of feedback, which we believe is based on a significant misunderstanding, is the concern about "minimal novelty" (auZ2, UQsK, BxBQ), particularly the claim that our work "duplicate[s] prior IC|TC work". **We must emphasize that our In-Context Clustering (ICC) paradigm is fundamentally different compared to IC|TC (Kwon et al., 2024).**
- IC|TC, as clarified in our paper (L105-124, L362-364, L374-377, L452-454), is a caption-based method. It first converts images into captions, and then uses LLMs to cluster those captions. This entire process is bottlenecked by the quality and relevance of the intermediate captions.
- **Our Method (ICC) doesn’t use captions at all.** We project image embeddings directly into the LLM's embedding space and the LLM then performs clustering by directly reasoning over these image embeddings in-context. This architecture is clearly illustrated in Figure 4 and described in L301-309.

We believe this fundamental misunderstanding of our core architecture led to an underestimation of our work's contribution. **Our paper is not restricted to being an image clustering method that leverages LLMs; instead, it studies the more fundamental problem of the LLM's capability to cluster data from diverse distributions in-context.** Specifically, our contributions are:
- **Zero-shot clustering capability**: We systematically evaluate the LLM's zero-shot clustering capability on numeric data from different distributions and compare different model families and sizes.
- **Enhance clustering capability through finetuning**: We use LoRA finetuning with NTP loss, a simple yet effective strategy that significantly improves clustering performance on numeric and image data.
- **Flexibility via text-conditioning**: By leveraging the flexibility of text prompting of LLMs, ICC can perform text-conditioned image clustering where the clustering criterion can be dynamically changed through the prompt. While a prior method IC|TC also studies this, it is constrained by a captioning bottleneck. We show how our method offers a cleaner solution and yields superior performance.
- **Novel analysis of attention structures**: Our work goes beyond simply using the final output and provides a novel analysis showing that meaningful cluster structures emerge directly within the LLM's attention matrices. We further demonstrate that these attention scores can be used directly with spectral clustering to achieve results that even outperform the model's generation.


We provided specific clarifications to each reviewer's concerns in the individual responses, with new baselines of classical methods.

---

### Meta-Review · Area_Chair_AxpA · 2026-01-05

**Summary:**

This paper proposes In-Context Clustering (ICC), a method that leverages Large Language Models (LLMs) to perform clustering across numerical and image data modalities, optionally enhanced via fine-tuning. The reviewers acknowledged several positive aspects: the paper is generally well-written and structured, and the analysis of attention maps to derive affinity matrices is an interesting exploratory insight. Some found the extension of in-context learning to unsupervised settings potentially significant. However, all reviewers raised major, converging weaknesses that undermine the paper’s contribution for ICLR:

Reviewers (auZ2, UQsK, BxBQ) unanimously found the core idea and technical pipeline to lack sufficient novelty. The work was perceived as closely following and only incrementally extending prior art, most notably the IC|TC framework for text-conditioned clustering, as well as other recent works on LLM-based and prompt-guided clustering (e.g., from CVPR, NeurIPS, EACL). Additionally, a critical flaw, emphasized by multiple reviewers (9K37, UQsK, BxBQ), is the incomplete and potentially unfair empirical comparison. The paper primarily compares against k-means and the IC|TC baseline, omitting comparisons with modern deep clustering methods and strong vision-feature baselines. These omissions make it impossible to assess whether the proposed LLM-based approach offers any practical advantage over established, often more efficient, methods. Furthermore, reviewers noted that comparisons with IC|TC may be confounded by the use of more capable base models (GPT-4o vs. GPT-3.5) or fine-tuning in the proposed method.

**Reviewer Concerns:**

The authors’ rebuttal attempted to address some concerns by clarifying related work distinctions and promising additional experiments. However, these responses were largely prospective and did not substantively alter the fundamental issues identified. The core criticisms regarding novelty, the scope of empirical validation, and methodological soundness remain unresolved.

**Reviewer Scores:**

It seems that the authors' rebuttal is difficult to provide substantive conceptual or methodological progress beyond the existing literature.

---

### Decision · Program_Chairs · 2026-01-26

Reject